# SGOOD: Substructure-enhanced Graph-Level Out-of-Distribution Detection

## Abstract

Graph-level representation learning is important in a wide range of applications. However, existing graph-level models are generally built on i.i.d. assumption for both training and testing graphs, which is not realistic in an open world, where models can encounter out-of-distribution (OOD) testing graphs that are from different distributions unknown during training. A trustworthy model should not only produce accurate predictions for in-distribution (ID) data, but also detect OOD graphs to avoid unreliable prediction. In this paper, we present SGOOD, a novel graph-level OOD detection framework. We find that substructure differences commonly exist between ID and OOD graphs. Hence, SGOOD effectively encodes task-agnostic substructures to learn powerful representations to achieve superior performance. Specifically, we build a super graph of substructures for every graph, and design a two-level graph encoding pipeline that works on both original graphs and super graphs to obtain substructure-enhanced graph representations. To further distinguish ID and OOD graphs, we develop three graph augmentation techniques that preserve substructures and increase expressiveness. Extensive experiments against 10 competitors on numerous graph datasets demonstrate the superiority of SGOOD, often surpassing existing methods by a significant margin. The code is available at `https://anonymous.4open.science/r/SGOOD-0958`.

## 1 Introduction

Graphs are ubiquitous to represent complex data, *e.g.*, chemical compounds, proteins, and social networks. Graph-level representation learning is crucial for applications in biochemistry (Jiang et al., 2021; Rong et al., 2020), social network (Dou et al., 2021; Shao et al., 2017), natural language processing (Peng et al., 2018; Xu et al., 2019), and recommendation (Wu et al., 2014).

Existing graph-level learning models are based on the closed-world assumption, in which testing graphs encountered at deployment are drawn i.i.d. from the same distribution as the training graph data. However, in reality, the models are actually in an *open world*, where testing graphs can be from different distributions that are never exposed to the models during training. In other words, testing graphs can be out-of-distribution (OOD) *w.r.t.* in-distribution (ID) training graphs (Li et al., 2022a;b; Yang et al., 2022). Consequently, the models trained by ID data tend to be inaccurate when making predictions on OOD data (Hendrycks & Gimpel, 2017), which raises reliability concerns in safety-critical applications, *e.g.*, drug discovery (Basile et al., 2019). A trustworthy graph-level learning model should not only give accurate predictions for ID graphs, but also determine whether a test graph is OOD or not, to avoid unreliable predictions.

Existing graph OOD detection methods, *e.g.*, (Li et al., 2022b; Liu et al., 2023b), mainly adopt message passing GNNs (Kipf & Welling, 2017; Hamilton et al., 2017) to first get node representations, and then generate graph-level representations solely based on these node representations. These methods do not consider the substructure patterns in graphs for OOD detection. In the literature, there are GNNs to learn high-order substructures in graphs, such as hierarchical GNNs (Ying et al., 2018; Lee et al., 2019; Gao & Ji, 2019) and subgraph GNNs (Zhao et al., 2021; Zhang & Li, 2021). These methods are trained using ID graphs and classification objectives to learn classification task-specific substructures. However, for OOD detection, OOD graphs are unseen during training, and thus these methods may achieve sub-optimal OOD detection performance (Winkens et al., 2020), as validated in our experiments. Winkens et al. (2020) propose that encoding the task-agnostic information into

representations can improve the OOD detection task on image data. Summing up, explicitly encoding task-agnostic substructures for graph-level OOD detection is underexplored in the literature.

In this paper, we develop SGOOD, a novel framework that explicitly encodes task-agnostic substructures and their relationships into effective representations for graph-level OOD detection. The design of SGOOD is motivated by the finding that *substructure differences of ID and OOD graphs commonly exist in real-world data*. We provide the following empirical evidence in Table 1. Given a dataset of graphs (see Table 2 for graph statistics), we apply modularity-based community detection (Clauset et al., 2004) to detect the substructures of the graphs. Note that the substructures are task-agnostic, since they are independent to specific learning tasks, *e.g.*, classification. Then we compute and report the percentage of OOD graphs with substructures that never appeared in ID graphs in Table 1. Observe that such percentage values are high, more than 44% in 4 out of 6 datasets, indicating many OOD graphs contain substructures rarely in ID graphs.

Table 1: The percentage of OOD graphs with substructures never appeared in ID graphs.

| Data | ENZYMES | IMDB-M | IMDB-B | BACE | BBBP | DrugOOD |
|------|---------|--------|--------|------|------|---------|
| | 58.9% | 14.0% | 8.5% | 50.0% | 44.6% | 77.3% |

The finding above justifies that, if one model can accurately preserve the substructures in ID graphs in an embedding space, intuitively, the model will give a large distance (i.e., OOD score) to OOD graphs with unseen substructures that are far away in the embedding space.

Hence, we develop a series of techniques in SGOOD to encode task-agnostic substructures and generate *substructure-enhanced graph representations* for effective graph-level OOD detection. Specifically, we first build a super graph $\mathcal{G}_i$ of substructures for every graph $G_i$ to obtain substructures and their relationships. Then, a *two-level graph encoding pipeline* is designed to work on $G_i$ and $\mathcal{G}_i$ in sequence to learn expressive substructure-enhanced graph representations. We prove that SGOOD with the pipeline is strictly more expressive than 1&2-WL, which theoretically justifies the power of preserving substructure patterns for OOD detection. To further enhance the performance, we design three *substructure-preserving graph augmentation* techniques. The augmentation techniques utilize the super graph of substructures to ensure that the substructures in a graph are modified as a whole. The overall training objective of SGOOD combines a classification loss with a contrastive loss. At test time, given a graph $G_i$ and its super graph $\mathcal{G}_i$, our OOD detector obtains the graph-level representations of both, which are then used for OOD score estimation. Extensive experiments are conducted to compare SGOOD against 10 baselines over many real-world graph datasets with various OOD types. SGOOD achieves superior performance, often outperforming existing methods by a significant margin. For instance, on an IMDB-M dataset, SGOOD achieves 9.58% absolute improvement in terms of AUROC over a runner-up baseline. In summary, our contributions are:

- We present SGOOD, a leading method that encodes task-agnostic substructures and their relationships to learn expressive representations for effective graph-level OOD detection.
- We design a novel two-level graph encoding pipeline by leveraging a constructed super graph of substructures, to empower SGOOD to learn substructure-enhanced graph representations.
- We further develop a collection of substructure-preserving graph augmentations via super graphs of substructures, to strengthen the distinguishability of SGOOD.
- Extensive experiments demonstrate the superiority of SGOOD for graph-level OOD detection.

## 2 PRELIMINARIES

We consider graph-level classification, which aims to classify a collection of graphs into different classes. Let $G_i = (V_i, E_i)$ be a graph, where $V_i$ and $E_i$ are node set and edge set, respectively. Let $\mathbf{x}_u \in \mathbb{R}^c$ denote the attribute vector of node $u \in V_i$ in graph $G_i$. Denote $\mathcal{X}$ as the in-distribution (ID) graph space and let $\mathcal{Y} = \{1, 2, ..., C\}$ be the label space. In graph-level classification, the training set $D_{tr}^{in} = \{(G_i, y_i)\}_{i=1}^n$ is drawn i.i.d. from the joint data distribution $P_{\mathcal{X}\mathcal{Y}}$. Every graph sample in $D_{tr}^{in}$ contains a graph $G_i$ with label $y_i$. Let $f$ be a learning model trained by the training set $D_{tr}^{in}$, and $f$ is deployed to predict the label of a testing graph.

**Graph-level Out-Of-Distribution Detection.** At test time, graph-level OOD detection can be treated a task to decide whether a testing graph $G_i$ in testing set $D_{test}$ is from the ID $P_{\mathcal{X}}$ or from other irrelevant distributions (*i.e.*, OOD). A typical way for OOD detection is to develop an OOD detector by leveraging the representations generated from the classification model $f$ that is trained

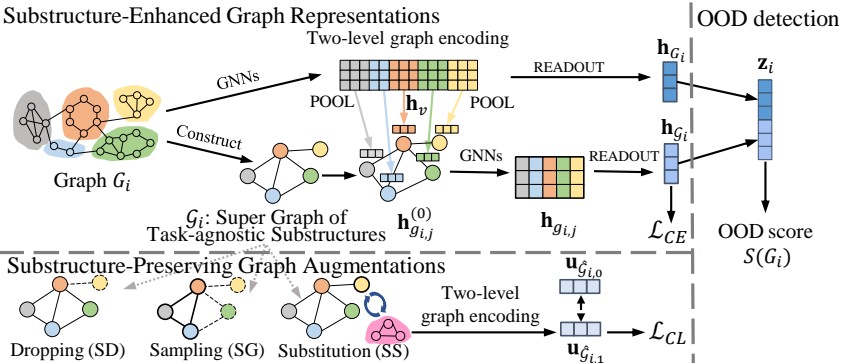

Figure 1: The SGOOD framework. The techniques in SGOOD are designed to effectively encode task-agnostic substructures and their relationships for accurate graph-level OOD detection.

via ID training graphs in $D_{tr}^{in}$. Specifically, the OOD detector has a scoring function $S(G_i)$ for every testing graph $G_i \in D_{test}$. Testing graphs with low scores $S(G_i)$ are regarded as ID, while the graphs with high scores are OOD. As stated in (Ming et al., 2023), a score threshold $\lambda$ is typically set so that a high fraction of ID data (*e.g.*, 95%) is correctly classified.

## 3 THE SGOOD METHOD

**Solution Overview.** The main goal of SGOOD is to effectively encode task-agnostic substructures and their relationships into representations for graph-level OOD detection. To achieve this, we develop several techniques in SGOOD as illustrated in Figure 1. Specifically, SGOOD generates *substructure-enhanced graph representations*, and further improves representation quality by *substructure-preserving graph augmentations*. Given a graph $G_i$, we first build its *super graph* $\mathcal{G}_i$ of task-agnostic substructures, in which a super node represents a substructure in $G_i$ and edges connect super nodes by following the connectivity in graph $G_i$. A *two-level graph encoding pipeline* is designed over both $G_i$ and $\mathcal{G}_i$ for graph-level representations that are enhanced by substructures. The training objective on ID training graphs is a cross-entropy loss $\mathcal{L}_{CE}$ for graph classification. For augmentations, intuitively, if more information about training ID data is preserved, it is easier to distinguish unseen OOD data. The substructure-preserving graph augmentations are designed to achieve this. Specifically, given a graph $G_i$, we augment it by first performing dropping, sampling, and substitution on its super graph $\mathcal{G}_i$ and then mapping the changes to $G_i$ accordingly. This process is substructure-preserving in the sense that a substructure is modified as a whole. The overall training objective of SGOOD combines a classification loss $\mathcal{L}_{CE}$ with a contrastive loss $\mathcal{L}_{CL}$. During test time, given a testing graph $G_i$, we first obtain the graph-level representations of both $G_i$ and its super graph $\mathcal{G}_i$, concatenate and normalize the representations, and finally get OOD score $S(G_i)$. The pseudo-code of SGOOD is in Appendix D.

### 3.1 SUBSTRUCTURE-ENHANCED GRAPH REPRESENTATIONS

As mentioned, substructures in a graph are critical to distinguish the graph from others. Given a graph $G_i$, we first describe how to construct its super graph $\mathcal{G}_i$ of substructures, and then present a two-level graph encoding pipeline to generate substructure-enhanced graph representations.

**Constructing a Super Graph of Substructures.** Let a substructure $g_{i,j}$ of graph $G_i$ be a connected subgraph of $G_i$. Specifically, a subgraph $g_{i,j} = (V_{i,j}, E_{i,j})$ is a substructure of $G_i = (V_i, E_i)$ iff $V_{i,j} \subseteq V_i$, $E_{i,j} \subseteq E_i$, and $g_{i,j}$ is connected. The substructures $\{g_{i,j}\}_{j=1}^{n_i}$ of a graph $G_i$ satisfy the following properties: (i) the node sets of substructures are non-overlapping, (ii) the union of the nodes in all substructures is the node set of $G_i$, and (iii) every substructure is a connected subgraph of $G_i$.

Remark that SGOOD uses task-agnostic substructures, and thus it is orthogonal to existing subgraph detection methods (Dhillon et al., 2007; Cordasco & Gargano, 2010), and it is not our focus on how to detect subgraphs. The substructures are detected without considering any learning tasks, e.g., classification, and thus they are task-agnostic. By default, we use modularity-based community detection (Clauset et al., 2004) to detect substructures. We also test other subgraph detection methods and find that the modularity-based substructures are effective in SGOOD, as shown in Table 6.

Then we construct the super graph $\mathcal{G}_i$ by regarding every substructure $g_{i,j}$ as a super node in $\mathcal{G}_i$, and connect super nodes by inserting edges via Definition 1. Super graph $\mathcal{G}_i$ can be regarded as a higher-order view that depicts the relationships between the substructures of a graph $G$. We also add self-loops in super graph $\mathcal{G}_i$.

**Definition 1** (A Super Graph of Substructures). A super graph of substructures constructed from the input graph $G_i = (V_i, E_i)$ is denoted as $\mathcal{G}_i = (\mathcal{V}_i, \mathcal{E}_i)$, where every super node $g_{i,j}$ in node set $\mathcal{V}_i = \{g_{i,j}\}_{j=1}^{n_i}$ represents a substructure of $G_i$, and every edge in $\mathcal{E}_i$ connecting two super nodes, and the edge set $\mathcal{E}_i = \{(g_{i,j}, g_{i,k}) | \exists u \in V_{i,j} \wedge v \in V_{i,k}, (u, v) \in E_i\}$.

**Two-level Graph Encoding.** Given a graph $G_i$ and its super graph $\mathcal{G}_i$, we present a two-level graph encoding pipeline, as shown in Figure 1. The idea is that, in addition to learning representations over $G_i$, we further utilize the super graph $\mathcal{G}_i$ to encode substructure information into graph representations, to better preserve distinguishable substructure patterns for effective OOD detection.

The two-level graph encoding first adopts GNNs to learn node representations with initial features over graph $G_i$. For every node $v \in V_i$, its representation $\mathbf{h}_v^{(l+1)}$ at $(l+1)$-layer is obtained by Eq. (1). Different GNNs have different AGGREGATE and COMBINE. By default, we adopt Graph Isomorphism Network (GIN) (Xu et al., 2018) as the backbone. The GIN for graph $G_i$ has $L_1$ layers.

$$\mathbf{h}_v^{(l+1)} = \text{COMBINE}^{(l+1)}(\mathbf{h}_v^{(l)}, \text{AGGREGATE}^{(l+1)}(\mathbf{h}_u^{(l)}, u \in N_{G_i}(v))), \mathbf{h}_v^{(0)} = \mathbf{x}_v, \qquad (1)$$

where $\mathbf{h}_v^{(l)} \in \mathbb{R}^d$ is the intermediate representation of node $v$ from the $l$-th layer GNNs with hidden dimension $d$, AGGREGATE$^{(l+1)}$ is the function that aggregates node features from $v$'s neighborhood $N_{G_i}(v)$ in graph $G_i$, COMBINE$^{(l+1)}$ is the function that updates node $v$'s representation by combining the representations of its neighbors with its own, and initially $\mathbf{h}_v^{(0)} = \mathbf{x}_v$.

Next, we obtain the representations of substructures $g_{i,j}$ in $G_i$ by leveraging the node representations above. As shown in Eq. (2), given a node $v$, we first concatenate all representations $\mathbf{h}_v^{(l)}$ for $l = 1, ..., L_1$ to get $\mathbf{h}_v$ that preserves multi-scale semantics. Then, for a substructure $g_{i,j}$ of graph $G_i$, we obtain the substructure representation $\mathbf{h}_{g_{i,j}}^{(0)}$ by integrating $\mathbf{h}_v$ of all $v$ in $g_{i,j}$ via DeepSet pooling (Zhang et al., 2019) in Eq. (2).

$$\mathbf{h}_{g_{i,j}}^{(0)} = \text{POOL}(\{\mathbf{h}_v | v \in V_{i,j}\}), \mathbf{h}_v = \text{CONCAT}(\{\mathbf{h}_v^{(l)}\}_{l=1}^{L_1}) \qquad (2)$$

Note that $\mathbf{h}_{g_{i,j}}^{(0)}$ only considers the nodes inside substructure $g_{i,j}$ and the original graph topology $G_i$. To further consider the relationships depicted in the super graph $\mathcal{G}_i$ of substructures, we regard $\mathbf{h}_{g_{i,j}}^{(0)}$ as the initial features of super node $g_{i,j}$ in $\mathcal{G}_i$, and employ a $L_2$-layer GIN over $\mathcal{G}_i$ to learn substructure-enhanced graph representations by Eq. (3) and (4).

$$\mathbf{h}_{g_{i,j}}^{(l+1)} = \text{COMBINE}^{(l+1)}(\mathbf{h}_{g_{i,j}}^{(l)}, \text{AGGREGATE}^{(l+1)}(\mathbf{h}_{g_{i,k}}^{(l)}, g_{i,k} \in N_{\mathcal{G}_i}(g_{i,j}))), \qquad (3)$$

where $N_{\mathcal{G}_i}(g_{i,j})$ contains the neighbors of super node $g_{i,j}$ in $\mathcal{G}_i$.

Lastly, in Eq. (4), we get the final representation $\mathbf{h}_{g_{i,j}}$ of every super node $g_{i,j}$ by concatenating the representation of $g_{i,j}$ in every layer of the $L_2$-layer GIN, and finally obtain the graph representation $\mathbf{h}_{\mathcal{G}_i}$ by a readout function that is sum pooling.

$$\mathbf{h}_{\mathcal{G}_i} = \text{READOUT}(\{\mathbf{h}_{g_{i,j}} | g_{i,j} \in \mathcal{V}_i\}), \mathbf{h}_{g_{i,j}} = \text{CONCAT}(\{\mathbf{h}_{g_{i,j}}^{(l)}\}_{l=0}^{L_2} \qquad (4)$$

Remark that the representation $\mathbf{h}_{\mathcal{G}_i}$ of super graph $\mathcal{G}_i$ of graph $G_i$ is used to train the loss $\mathcal{L}_{CE}$ for classification. Meanwhile, as explained shortly, for OOD detection during testing, we further consider another representation of graph $G_i$ obtained by aggregating node representations as in Figure 1.

**Discussion.** In literature, there exist studies considering substructures/subgraphs for graph representation learning, such as hierarchical pooling (Ying et al., 2018; Lee et al., 2019; Gao & Ji, 2019) and subgraph GNNs (Zhang & Li, 2021; Zhao et al., 2021). We also conduct experiments to demonstrate that our SGOOD is more effective than these methods for the task of graph-level OOD detection.

### 3.2 SUBSTRUCTURE-PRESERVING GRAPH AUGMENTATIONS

Intuitively, if more information about training ID data is preserved, it is easier to distinguish unseen OOD data. Hence, we design substructure-preserving graph augmentations by leveraging the super

graph $\mathcal{G}_i$ of graph $G_i$, to improve the performance further. However, it is challenging to achieve this. Substructures with subtle differences have different semantics. It is important to keep the substructures of a graph intact while performing augmentations. Common augmentation techniques like edge permutation and node dropping directly on graphs $G_i$ may unexpectedly destroy meaningful substructures, and hamper OOD detection effectiveness.

To tackle the issue, we first perform augmentations on the super graph $\mathcal{G}_i$ by regarding substructures as atomic nodes, and then map the augmentations to the original graph $G_i$ with modifications over substructures as a whole. Specifically, we propose three substructure-level graph augmentations below, namely *substructure dropping (SD), super graph sampling (SG), and substructure substitution (SS)*. The default augmentation ratio is set to 0.3.

- *Substructure Dropping (SD)*. Given a graph $G_i$ with its super graph $\mathcal{G}_i$, a fraction of super nodes in $\mathcal{G}_i$ (*i.e.*, the corresponding substructures in $G_i$) are discarded uniform randomly. Remark that selected substructures are dropped as a whole.
- *Super Graph Sampling (SG)*. In the super graph $\mathcal{G}_i$, we start from a random node, sample a fixed-size connected subgraph in $\mathcal{G}_i$, and drop the rest nodes and edges. The changes are mapped to $G_i$ accordingly. Depth-first search is chosen as the sampling strategy (You et al., 2020).
- *Substructure Substitution (SS)*. Given a graph $G_i$ in class $c$ with super graph $\mathcal{G}_i$ of substructures, we randomly substitute a fraction of nodes in $\mathcal{G}_i$ (*i.e.*, substructures in $G_i$) with other substructures from the graphs of the same class $c$. To avoid drastic semantic change of the whole graph, only super nodes with degree one (excluding self-loops) in $\mathcal{G}_i$ take part in the substitution.

### 3.3 OBJECTIVES AND TRAINING

For classification, we adopt a standard cross-entropy loss $\mathcal{L}_{CE}$. Specifically, after getting the representation $\mathbf{h}_{\mathcal{G}_i}$ for the super graph $\mathcal{G}_i$ of graph $G_i$, we apply a linear transformation to get prediction logits $\widehat{y}_i$, which is evaluated against the ground-truth class label $y_i$ to get $\mathcal{L}_{CE}$ by Eq. (5) for a mini-batch of $B$ training graphs.

$$\mathcal{L}_{CE} = -\frac{1}{B} \sum_{i=1}^{B} \sum_{c=1}^{C} \mathbb{1}(y_i = c) \log (\hat{y}_{i,c}) \qquad (5)$$

Then we adopt the substructure-preserving graph augmentations in Section 3.2 to get contrastive loss $\mathcal{L}_{CL}$. Specifically, given a mini-batch of $B$ training graphs $\{G_i\}_{i=1}^{B}$ and their super graphs $\{\mathcal{G}_i\}_{i=1}^{B}$, we transform the super graphs to get $\widehat{\mathcal{G}}_{i,0} = \mathcal{T}_0(\mathcal{G}_i)$ and $\widehat{\mathcal{G}}_{i,1} = \mathcal{T}_1(\mathcal{G}_i)$, where $\mathcal{T}_0$ and $\mathcal{T}_1$ are two augmentations chosen among $\mathcal{A} = \{I, SD, SG, SS\}$, where $I$ indicates no augmentation. Graph $G_i$ is transformed accordingly via $\mathcal{T}_0$ and $\mathcal{T}_1$ to obtain $\widehat{G}_{i,0}$ and $\widehat{G}_{i,1}$ respectively. Then, the representations $\mathbf{h}_{\widehat{\mathcal{G}}_{i,0}}$ and $\mathbf{h}_{\widehat{\mathcal{G}}_{i,1}}$ of the two augmented super graphs can be calculated by applying Eq.(1)-(4). We transform $\mathbf{h}_{\widehat{\mathcal{G}}_{i,0}}$ and $\mathbf{h}_{\widehat{\mathcal{G}}_{i,1}}$ by a shared projection head $\psi(\cdot)$, which is a 2-layer MLP, followed by $l_2$-normalization, to obtain $\mathbf{u}_{\widehat{\mathcal{G}}_{i,0}} = \psi(\mathbf{h}_{\widehat{\mathcal{G}}_{i,0}})/||\psi(\mathbf{h}_{\widehat{\mathcal{G}}_{i,0}})||$ and $\mathbf{u}_{\widehat{\mathcal{G}}_{i,1}} = \psi(\mathbf{h}_{\widehat{\mathcal{G}}_{i,1}})/||\psi(\mathbf{h}_{\widehat{\mathcal{G}}_{i,1}})||$, respectively. We get $\mathcal{L}_{CL}$ by

$$\mathcal{L}_{CL} = \frac{1}{2B} \sum_{i=1}^{B} \sum_{a \in \{0,1\}} -\log \frac{\exp (\mathbf{u}_{\widehat{\mathcal{G}}_{i,a}}^{\intercal} \mathbf{u}_{\widehat{\mathcal{G}}_{i,1-a}}/\tau)}{\sum_{j=1}^{B} \exp (\mathbf{u}_{\widehat{\mathcal{G}}_{i,a}}^{\intercal} \mathbf{u}_{\widehat{\mathcal{G}}_{j,1-a}}/\tau) + \sum_{j=1}^{B} \mathbb{1}(j \neq i) \exp (\mathbf{u}_{\widehat{\mathcal{G}}_{i,a}}^{\intercal} \mathbf{u}_{\widehat{\mathcal{G}}_{j,a}}/\tau)}, \qquad (6)$$

where $\tau$ is a temperature parameter.

The overall training loss is

$$\mathcal{L} = \mathcal{L}_{CE} + \alpha \mathcal{L}_{CL}, \text{ where } \alpha \text{ is a weight factor.} \qquad (7)$$

The training procedure of SGOOD consists of two stages. In the first pre-training stage, the parameters are solely updated by minimizing $\mathcal{L}_{CL}$ for $T_{PT}$ epochs. In the second stage, the parameters are fine-tuned under the combined overall loss $\mathcal{L}$ for $T_{FT}$ epochs. This training procedure achieves better performance than directly training $\mathcal{L}$, as shown in Appendix Figure 4 when pretraining $T_{PT}$ is 0.

### 3.4 GRAPH-LEVEL OOD SCORING

Recall that the main goal of OOD detection is to let the representations of ID data and OOD data to be distant from each other. In terms of distance, at test time, given a testing graph $G_i \in D_{test}$, we

Table 2: Data Statistics.

| Dataset | Graph Type | OOD Type | # Class | # ID Train | # ID Val | # ID Test | # OOD Test |
|---|---|---|---|---|---|---|---|
| ENZYMES (Morris et al., 2020) | Proteins | Unseen Classes | 6 | 480 | 60 | 60 | 60 |
| IMDB-M (Morris et al., 2020) | Social Networks | Unseen Classes | 3 | 1200 | 150 | 150 | 150 |
| IMDB-B (Morris et al., 2020) | Social Networks | Unseen Classes | 2 | 800 | 100 | 100 | 100 |
| REDDIT-12K (Yanardag & Vishwanathan, 2015) | Social Networks | Unseen Classes | 11 | 6997 | 875 | 875 | 875 |
| BACE (Wu et al., 2018) | Molecules | Scaffold | 2 | 968 | 121 | 121 | 121 |
| BBBP (Wu et al., 2018) | Molecules | Scaffold | 2 | 1303 | 164 | 164 | 164 |
| DrugOOD (Ji et al., 2022) | Molecules | Protein Target | 2 | 800 | 100 | 100 | 100 |
| HIV (Wu et al., 2018) | Molecules | Scaffold | 2 | 26319 | 3291 | 3291 | 3291 |

use the standard Mahalanobis distance (Lee et al., 2018) to quantify its OOD score. If $G_i$ is with large Mahalanobis distance from the ID training data in the embedding space, it tends to be OOD.

In SGOOD shown in Figure 1, in addition to the representation $\mathbf{h}_{\mathcal{G}_i}$ of the super graph $\mathcal{G}_i$ of a testing graph $G_i$, we also aggregate the node representations of $G_i$ to get $\mathbf{h}_{G_i} = \texttt{READOUT}(\{\mathbf{h}_v | v \in V_i\})$. Representations $\mathbf{h}_{\mathcal{G}_i}$ and $\mathbf{h}_{G_i}$ are concatenated together to estimate the OOD score $S(G_i)$

$$S(G_i) = \max_{c \in [C]} (\mathbf{z}_i - \boldsymbol{\mu}_c)^\intercal \widehat{\Sigma}^{-1} (\mathbf{z}_i - \boldsymbol{\mu}_c), \tag{8}$$

$$\boldsymbol{\mu}_c = \frac{1}{N_c} \sum_{j:y_j=c} \mathbf{z}_j; \widehat{\Sigma} = \frac{1}{N} \sum_{c \in [C]} \sum_{j:y_j=c} (\mathbf{z}_j - \boldsymbol{\mu}_c)(\mathbf{z}_j - \boldsymbol{\mu}_c)^\intercal; \mathbf{z}_i = \frac{\texttt{CONCAT}(\mathbf{h}_{G_i}, \mathbf{h}_{\mathcal{G}_i})}{||\texttt{CONCAT}(\mathbf{h}_{G_i}, \mathbf{h}_{\mathcal{G}_i})||_2}, \tag{9}$$

where $[C] = \{1, 2, \ldots, C\}$, $\boldsymbol{\mu}_c$ is the estimated class centroid for class $c$, and $\widehat{\Sigma}$ is the estimated covariance matrix for ID graphs.

### 3.5 ANALYSIS

We show in Proposition 1 that SGOOD is more expressive than 1&2-WL, indicating that SGOOD can distinguish more structural patterns, which, together with our empirical findings in Section 1, explains the power of SGOOD for graph-level OOD detection. The proof is provided in Appendix A.

**Proposition 1.** When the GNNs adopted in SGOOD are with sufficient number of layers, and the POOL function in Eq.(2) and READOUT function in Eq.(4) are injective, then SGOOD is strictly more powerful than 1&2-WL.

## 4 EXPERIMENTS

**Datasets and Evaluation Metrics.** We adopt real datasets that encompass diverse types of OOD graphs, as listed in Table 2. The OOD graph data is generated following (Liu et al., 2023b; Li et al., 2022b). All ID graphs $D^{in}$ are randomly split into training, validation, and testing with ratio 8:1:1, following the settings of standard graph classification (Hu et al., 2020; Morris et al., 2020). The testing set consists of the same number of ID graphs and OOD graphs. We use three commonly used metrics AUROC, AUPR and FPR95 for OOD detection evaluation (Hendrycks & Gimpel, 2017; Wu et al., 2022). All these metrics are independent of threshold choosing. For the classification performance in ID graphs, we use Accuracy (ID ACC). Remark that the priority of the graph-level OOD detection task is to accurately identify OOD graphs, when maintaining instead of improving the ID ACC. The details of dataset descriptions and the metric formula are provided in Appendix B.1.

**Baselines and Implementation Details.** We compare SGOOD with 10 competitors in 3 categories. (i) General OOD detection methods, including *MSP* (Hendrycks & Gimpel, 2017), *Energy* (Liu et al., 2020), *ODIN* (Liang et al., 2018), and *MAH* (Lee et al., 2018), for each of which, we replace their network backbone with GIN to handle graph data. (ii) Existing graph-level OOD detection methods, including *GNNSafe* (Wu et al., 2022), *GraphDE* (Li et al., 2022b) and *GOOD-D* (Liu et al., 2023b). (iii) Existing graph-level anomaly detection methods, including *OCGIN*(Zhao & Akoglu, 2021), *OCGTL*(Qiu et al., 2022), and *GLocalKD*(Ma et al., 2022). In SGOOD, we set the number of layers $L_1 = 3$ and $L_2 = 2$, and dimension $d$ as 16. We use mini-batch gradient descent to optimize parameters in SGOOD with Adam optimizer, and batch size is set as 128. In SGOOD, we set $T_{PT}$ as 100 epochs and $T_{FT}$ as 500 epochs. In the first stage of training, learning rate is tuned in $\{0.01, 0.001, 0.0001\}$. For the second stage, we set learning rate as 0.001 and $\alpha$ as 0.1. On each dataset, we repeat experiments 5 times with different random seeds and report the mean metrics. The implementation details of baselines are in Appendix B.3.

Table 3: Overall OOD detection performance by AUROC, AUPR, and FPR95 in percentage % (mean ± std). ↑ indicates larger values are better and vice versa. **Bold**: best. Underline: runner-up.

| Method | ENZYMES AUROC↑ | AUPR↑ | FPR95↓ | IMDB-M AUROC↑ | AUPR↑ | FPR95↓ | IMDB-B AUROC↑ | AUPR↑ | FPR95↓ | REDDIT-12K AUROC↑ | AUPR↑ | FPR95↓ |
|---|---|---|---|---|---|---|---|---|---|---|---|---|
| MSP | 61.34±3.79 | 61.65±6.64 | 89.67±2.26 | 42.75±1.52 | 51.04±1.93 | 95.73±1.63 | 58.13±2.31 | 59.63±1.22 | 91.40±4.16 | 50.63±0.87 | 48.60±1.08 | 95.95±1.25 |
| Energy | 54.69±9.18 | 56.90±8.85 | 89.33±3.55 | 24.50±19.73 | 37.26±11.78 | 96.40±2.25 | 49.58±17.76 | 59.03±13.06 | 92.80±3.55 | 55.10±0.48 | 56.52±0.78 | 97.19±0.58 |
| ODIN | 63.70±2.70 | 65.72±4.77 | 92.66±3.26 | 40.12±2.96 | 50.08±2.44 | 96.66±1.03 | 58.25±2.94 | 61.36±0.49 | 92.20±2.92 | 51.74±2.03 | 54.53±1.26 | 96.45±0.73 |
| MAH | 67.37±3.67 | 63.81±2.15 | 83.33±9.60 | 69.26±3.67 | 63.64±2.14 | 60.93±9.06 | 76.77±4.37 | 76.88±6.30 | 81.40±7.14 | 72.68±0.87 | 74.47±0.48 | 80.75±2.05 |
| GNNSafe | 56.85±8.91 | 56.13±8.26 | 97.00±3.71 | 21.93±1.76 | 36.88±1.68 | 95.46±1.42 | 70.49±14.80 | 75.67±15.71 | 87.80±5.81 | 51.68±0.08 | 53.97±0.52 | 95.59±2.80 |
| GraphDE | 61.35±3.99 | 66.26±2.98 | 99.00±0.81 | 66.87±4.25 | 62.60±4.47 | 93.06±8.24 | 26.91±3.35 | 42.73±2.06 | 100.00±0.00 | 59.40±0.18 | 63.06±0.30 | 81.82±0.01 |
| GOOD-D | 67.21±6.41 | 64.86±6.32 | 82.33±8.31 | 61.89±4.87 | 66.91±7.60 | 95.20±4.55 | 52.58±10.21 | 55.69±10.56 | 99.20±1.00 | 56.11±0.10 | 59.56±0.16 | 93.67±0.34 |
| OCGIN | 68.11±4.61 | 68.90±4.19 | 89.67±3.70 | 47.51±9.47 | 50.76±4.53 | 98.27±17.70 | 60.78±5.21 | 57.80±5.10 | 8780±9.15 | 59.33±1.26 | 60.02±1.88 | 90.00±2.01 |
| GLocalKD | 71.46±3.21 | 64.93±4.44 | 78.67±6.37 | 19.82±1.57 | 35.39±0.49 | 98.27±1.13 | 79.39±4.71 | **85.56±3.33** | 87.40±5.42 | 49.60±1.06 | 51.75±0.72 | 97.60±0.35 |
| OGGTL | 73.22±1.83 | **73.61±3.19** | 82.33±2.70 | 54.07±12.93 | 58.20±7.86 | 86.40±6.49 | 37.39±18.82 | 47.11±14.06 | 98.80±2.40 | 51.62±0.019 | 53.33±0.01 | 96.79±0.06 |
| SGOOD | **74.40±1.42** | 72.53±2.51 | **73.66±7.03** | **78.84±2.00** | **72.54±3.21** | **45.46±6.62** | **80.41±3.16** | 83.49±3.59 | **81.20±2.28** | **74.95±0.79** | **74.93±0.93** | **75.17±2.72** |

| Method | BACE AUROC↑ | AUPR↑ | FPR95↓ | BBBP AUROC↑ | AUPR↑ | FPR95↓ | DrugOOD AUROC↑ | AUPR↑ | FPR95↓ | HIV AUROC↑ | AUPR↑ | FPR95↓ |
|---|---|---|---|---|---|---|---|---|---|---|---|---|
| MSP | 46.34±6.10 | 48.65±3.08 | 97.02±2.18 | 57.37±4.28 | 56.84±3.36 | 94.63±2.26 | 52.86±5.26 | 54.49±4.33 | 98.80±0.01 | 50.75±1.88 | 50.49±0.91 | 95.52±0.50 |
| Energy | 46.05±6.66 | 49.68±4.16 | 97.36±2.92 | 56.56±4.16 | 55.74±2.78 | 92.68±2.62 | 52.81±5.36 | 54.98±4.36 | 98.20±1.16 | 50.97±2.13 | 50.49±0.91 | 95.50±0.59 |
| ODIN | 45.51±3.85 | 48.28±3.76 | 97.02±1.53 | 54.78±3.46 | 54.63±3.69 | 96.34±1.80 | 51.09±3.79 | 52.70±2.66 | 99.00±1.09 | 50.16±0.73 | 49.95±0.58 | 94.60±1.07 |
| MAH | 73.78±1.97 | 75.33±2.32 | 86.78±6.32 | 53.77±4.27 | 52.57±3.81 | 93.29±2.51 | 66.90±4.14 | 64.30±4.43 | 81.60±4.58 | 58.10±3.60 | 57.18±3.18 | 91.89±1.32 |
| GNNSafe | 47.61±7.50 | 51.52±5.91 | 98.18±2.05 | 47.04±2.40 | 51.52±5.90 | 98.41±0.99 | 50.44±0.57 | 51.14±0.30 | 96.01±0.33 | 50.98±6.82 | 55.13±6.81 | 96.01±0.33 |
| GraphDE | 47.32±1.52 | 51.1±2.57 | 94.21±4.58 | 50.88±2.78 | 51.47±3.84 | 94.63±2.34 | 60.19±4.32 | 62.59±2.47 | 88.80±5.60 | 52.38±1.86 | 54.14±3.21 | 94.89±0.84 |
| GOOD-D | 70.42±2.22 | 73.21±3.34 | 88.26±1.78 | 54.15±1.10 | 58.58±1.93 | 99.39±0.41 | 60.52±3.33 | 63.09±2.54 | 98.40±1.27 | 59.69±0.62 | 57.10±.14 | 92.03±0.61 |
| OCGIN | 59.71±5.20 | 61.43±5.18 | 93.39±4.44 | 47.78±5.72 | 47.27±2.98 | 94.76±2.70 | 57.95±5.80 | 59.50±7.00 | 94.20±3.12 | 54.06±0.47 | 52.14±0.26 | 92.81±1.01 |
| GLocalKD | 45.34±2.11 | 55.39±2.35 | 98.68±1.11 | 43.77±2.23 | 45.84±1.20 | 98.29±1.00 | 45.72±0.97 | 50.90±3.33 | 100.00±0.00 | 46.81±2.90 | 46.95±2.01 | 97.05±0.19 |
| OGGTL | 80.84±2.00 | 79.93±1.26 | 66.44±8.89 | 58.73±2.19 | **60.47±1.38** | 91.46±2.21 | 67.59±7.93 | 70.90±5.80 | 83.00±11.22 | 51.78±0.19 | 53.71±0.22 | 96.41±0.05 |
| SGOOD | **84.39±2.73** | **83.32±2.49** | **64.13±4.83** | **61.25±1.60** | 59.36±2.39 | **88.04±3.44** | **73.15±4.48** | **73.25±4.49** | **67.40±5.16** | **60.82±0.75** | **59.99±0.69** | **90.39±1.04** |

## 4.1 OVERALL GRAPH-LEVEL OOD DETECTION EFFECTIVENESS

Table 3 reports the overall graph-level OOD detection performance of all methods by AUROC, AUPR and FPR95 metrics on all datasets, by mean and standard deviation values. Observe that SGOOD consistently achieves superior OOD detection effectiveness under most settings. For instance, on IMDB-M, SGOOD has AUROC 78.84%, which indicates $9.58\%$ absolute improvement over the best competitor with AUROC 69.26%. As another example on BACE molecule dataset, the AUROC of SGOOD is 84.39%, while the runner-up achieves AUROC 80.84%. The overall results in Table 3 demonstrate the power of our technical designs in SGOOD presented in Section 3 for graph-level OOD detection. Due to space limit, ID ACC results are in Appendix Table 9.

## 4.2 MODEL ANALYSIS

**Ablation.** In Table 4, SGOOD\A is SGOOD that ablates all augmentations in Section 3.2, *i.e.*, $\alpha$=0 in Eq. (7); SGOOD (base) further ablates all substructure-related representations in Section 3.1. In Table 4, first observe that, from SGOOD (base) to SGOOD\A and then to the complete version SGOOD, the performance gradually increases on all datasets, validating the effectiveness of all proposed techniques. Second, SGOOD\A already surpasses the best baseline performance on most datasets, which demonstrates the effect of the techniques in Section 3.1, without the augmentation techniques in Section 3.2. With the help of the substructure-preserving graph augmentations, SGOOD pushes the performance further higher.

Table 4: Ablation AUROC (%)

| Method | ENZYMES | IMDB-M | IMDB-B | BACE | BBBP | DrugOOD |
|---|---|---|---|---|---|---|
| Best baseline | 71.46 | 69.26 | 79.39 | 73.78 | 57.37 | 57.37 |
| SGOOD (base) | 67.38 | 69.26 | 76.80 | 73.78 | 53.77 | 66.90 |
| SGOOD\A | 73.60 | 75.22 | 77.80 | 75.96 | 57.84 | 68.80 |
| SGOOD | **74.41** | **78.84** | **80.42** | **84.40** | **61.25** | **73.16** |

Figure 2: ID and OOD score distributions with the dotted line indicating the mean of ID/OOD scores.

In Figure 2, we visualize the ID and OOD score distributions of SGOOD (base), SGOOD\A and SGOOD on DrugOOD, with their mean scores shown in dotted lines. Clearly, we are obtaining more

separable OOD scores against ID data from left to right in Figure 2, which demonstrates that our techniques in SGOOD can learn distinguishable representations for ID and OOD graphs.

**Effect of Substructure-Preserving Graph Augmentations.** We evaluate the augmentations (SD, SG, and SS) in Section 3.2, with conventional graph augmentations that are not substructure-preserving, including edge perturbation (EP), attribute masking (AM), node dropping (ND), and subgraph sampling (SA). Table 5 reports the results, AM is not applicable on IMDB-M and IMDB-B since they do not have node attributes. Observe that our SD, SG, and SS are the top-3 ranked techniques for graph-level OOD detection, validating the effectiveness of the proposed substructure-preserving graph augmentations. In Appendix Figure 7, we also visualize the improvements of all pairwise combinations of our augmentation techniques.

Table 5: Comparing with different augmentations by AUROC (%)

|      | ENZYMES | IMDB-M | IMDB-B | BACE  | BBBP  | DrugOOD | Avg. Rank |
|------|---------|--------|--------|-------|-------|---------|-----------|
| EP   | 74.28   | 76.50  | 78.44  | 78.75 | 58.24 | 71.32   | 4.67      |
| AM   | 72.44   | /      | /      | 77.28 | 59.68 | 71.27   | 5.25      |
| ND   | 73.11   | 77.09  | 78.40  | 78.79 | 58.59 | 69.48   | 4.83      |
| SA   | 72.12   | 76.76  | 79.25  | 77.13 | 57.84 | **72.66** | 4.83    |
| SD   | **74.77** | **78.15** | 79.54 | 82.00 | 59.76 | 72.65 | **1.83**  |
| SG   | 72.74   | 77.98  | 78.97  | 82.24 | 59.58 | 71.97   | 3.33      |
| SS   | 74.27   | 76.20  | **80.50** | **83.53** | **63.53** | 71.94 | 2.67 |

**Effect of Substructure Extraction Methods.** As mentioned, SGOOD is orthogonal to specific substructure extraction methods. Here in SGOOD, we evaluate several commonly-used methods to extract substructures, including Graclus (Dhillon et al., 2007), label propagation (LP) (Cordasco & Gargano, 2010), and BRICS (Degen et al., 2008). Specifically, BRICS uses Chemistry knowledge for extraction. In Table 6, SGOOD with different substructure detections are all better than SGOOD *w.o.* using substructures, and SGOOD with Modularity is the best. The results validate the effectiveness of our framework that leverages substructures for OOD detection.

Table 6: Comparison between different substructure detection methods by AUROC (%).

| SGOOD | ENZYMES | IMDB-M | IMDB-B | BACE  | BBBP  | DrugOOD |
|-------|---------|--------|--------|-------|-------|---------|
| *w.o.* substructures | 67.38 | 69.26 | 76.8 | 73.78 | 53.77 | 66.90 |
| Modularity | **74.41** | **78.84** | **80.42** | **84.40** | **61.25** | **73.16** |
| Graclus | 71.12 | 74.64 | 78.86 | 79.54 | 56.62 | 67.94 |
| LP | 68.09 | 75.48 | 78.46 | 76.63 | 54.90 | 68.95 |
| BRICS | / | / | / | 78.39 | 60.18 | 64.78 |

**Comparison with Subgraph-aware Models.** We then compare SGOOD directly with subgraph-aware models, including three hierarchical pooling methods (SAG (Lee et al., 2019), TopK (Gao & Ji, 2019), DiffPool (Ying et al., 2018)) and two subgraph GNNs (NGNN (Zhang & Li, 2021) and GNN-AK$^+$ (Zhao et al., 2021)). Note that these methods are not specifically designed for graph-level OOD detection. At test time, we extract the graph representations generated by these methods and use Mahalanobis distance as OOD score. Table 7 reports the AUROC results. SGOOD performs best on 5 out of 6 datasets and is the top-2 on DrugOOD. This validates the effectiveness of our substructure-related techniques in Section 3 for graph-level OOD detection.

Table 7: Comparing with subgraph-aware models AUROC (%). **Bold**: best. Underline: runner-up.

| Method | ENZYMES | IMDB-M | IMDB-B | BACE | BBBP | DrugOOD |
|--------|---------|--------|--------|------|------|---------|
| SAG | 70.40 | 76.50 | 77.30 | 76.90 | 58.90 | 65.80 |
| TopK | 70.20 | 76.80 | 77.20 | 74.20 | 54.90 | 58.50 |
| DiffPool | 73.30 | 75.90 | 78.00 | 76.50 | 57.50 | 70.60 |
| NGNN | 70.30 | 71.20 | 76.60 | 71.20 | 52.60 | **75.60** |
| GNN-AK$^+$ | 68.50 | 73.50 | 77.20 | 70.90 | 54.30 | 63.00 |
| SGOOD | **74.41** | **78.84** | **80.42** | **84.40** | **61.25** | 73.16 |

**Varying $L_1$ and $L2$.** In the experiments above, we fix the layers of the two GINs in the two-level graph encoding in Section 3.1 to be $L_1 = 3$ and $L_2 = 2$ as default. If we search $L_1$ and $L_2$, it is possible to get even better OOD detection results, as shown in Table 8 where $L_1$ and $L_2$ are varied with their sum fixed to be 5. For example, on BACE with $L_1$=2 and $L_2$=3, AUROC is 62.26%, about 1% higher than the default setting.

Table 8: Varying $L_1$ and $L_2$ in SGOOD (AUROC).

| $L_1$ | $L_2$ | ENZYMES | IMDB-M | IMDB-B | BBBP | BACE | DrugOOD |
|-------|-------|---------|--------|--------|------|------|---------|
| 4 | 1 | 74.00 | 77.13 | **81.00** | 80.43 | 62.00 | 71.17 |
| 3 | 2 | **74.41** | **78.84** | 80.42 | 84.40 | 61.25 | **73.16** |
| 2 | 3 | 73.63 | 76.03 | 79.05 | 80.34 | **62.26** | 69.12 |
| 1 | 4 | 74.22 | 77.83 | 76.79 | 76.62 | 61.08 | 68.01 |

**Efficiency.** The training and inference time is in Appendix Table 11. SGOOD is faster to train than existing graph-level OOD detection methods, including GraphDE and GOOD-D, and has close efficiency with OCGIN and GLocalKD. SGOOD also has close training time to general OOD detection methods. All methods have close inference time.

**More experiments.** In Appendix, we compare the performance of SGOOD and baselines when different backbones other than GIN are used in Table 10, evaluate the effect when varying augmentation ratio in Figure 3, study the effect of pretraining epochs $T_{PT}$ in Figure 4, vary the weight of constrastive loss $\alpha$ in Figure 5, and visualize detected substructures in Figure 8.

## 5 RELATED WORK

**Graph-level Representation Learning.** Graph-level representation learning aims to learn representations of entire graphs (Wu et al., 2020). GNNs (Hamilton et al., 2017; Kipf & Welling, 2017; Veličković et al., 2018; Xu et al., 2018) are often adopted (Guo et al., 2023; Yang et al., 2022) to first learn node representations by message passing on graphs and then node representations are aggregated by pooling functions to get graph-level representations (Liu et al., 2023a). It is shown that the expressiveness of such a way is limited by 1-WL (Chen et al., 2020; Li et al., 2020). Hence, there are general subgraph-aware methods to improve the expressive power, *e.g.*, hierarchical pooling (Gao & Ji, 2019; Lee et al., 2019; Ying et al., 2018) and subgraph GNNs (Zhang & Li, 2021; Zhao et al., 2021). Hierarchical pooling methods learn to assign nodes into different clusters and coarsen graphs hierarchically. Subgraph GNNs apply message passing on extracted rooted-subgraphs of nodes in a graph, and then aggregate subgraph representations (Frasca et al., 2022). Wang et al. (2022) assumes pre-defined repetitive type of substructures on periodic graphs. Contrarily, we do not have such an assumption and consider various substructures automatically extracted by community detection. Still, these methods assume that graphs are i.i.d in training and testing. In experiments, SGOOD is more effective for graph-level OOD detection.

**Out-of-distribution Detection.** OOD detection has received great research attention in various data domains, as learning models tend to be over-confident on out-of-distribution data (Hendrycks & Gimpel, 2017; Nguyen et al., 2015). There are OOD detection methods designed for image data (Hendrycks & Gimpel, 2017; Lee et al., 2018; Liang et al., 2018; Liu et al., 2020; Ming et al., 2023; Sehwag et al., 2021; Sun et al., 2022). Some methods rely on classification probabilities predicted by neural networks to get OOD scores (Hendrycks & Gimpel, 2017; Liang et al., 2018; Liu et al., 2020), while the others measure OOD scores according to the distance between test samples and ID training data (Lee et al., 2018; Ming et al., 2023; Sehwag et al., 2021; Sun et al., 2022). Note that non-graph OOD detection methods are designed without considering the unique characteristics of graphs, though they can be modified to handle OOD detection on graph data. As shown in our experiments, SGOOD surpasses these methods. Recently, several OOD detection methods on graphs have been proposed. Wu et al. (2022) explore node-level OOD detection by using energy function to detect OOD nodes in a graph, which is a different problem from this paper. For graph-level OOD detection, Li et al. (2022b) design a generative model that has the ability to identify outliers in training graph samples, as well as OOD samples during the testing stage. Liu et al. (2023b) develop a self-supervised learning approach to train their model to estimate OOD scores at test time. Recently, Zhang et al. (2022) proposes to learn anomalous substructures using deep random walk kernel, which depends on labeled anomalous graphs, while OOD graphs are unseen during the training stage and only available during the testing stage. Observe that existing graph-level OOD detection methods mainly leverage node representations output by GNNs (Kipf & Welling, 2017; Veličković et al., 2018; Xu et al., 2018) to get graph-level representations, while the rich substructure patterns hidden in graphs are under-investigated for graph-level OOD detection. On the other hand, our method SGOOD explicitly uses substructures in graphs to learn high-quality graph-level representations for effective graph-level OOD detection. The representations generated by SGOOD are able to better distinguish ID and OOD graphs than existing methods, which have already been demonstrated in the extensive experiments.

## 6 CONCLUSION

We study the problem of graph-level OOD detection, and present a novel SGOOD method with superior performance. The design of SGOOD is motivated by the exciting finding that substructure differences commonly exist between ID and OOD graphs. By leveraging substructures, SGOOD aims to preserve more distinguishable graph-level representations between ID and OOD graphs. Specifically, we build a super graph of substructures for every graph, and develop a two-level graph encoding pipeline to obtain high-quality structure-enhanced graph representations. We theoretically prove the expressiveness power of the obtained representations. To further improve the representation quality, we develop a set of substructure-preserving graph augmentations. Extensive experiments on real-world graph datasets validate the superior performance of SGOOD over existing methods for graph-level OOD detection.

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

APPENDIX

We provide the proof of Proposition 1 in Appendix A, more experimental details on datasets, evaluation metrics, implementation in Appendix B, several additional experiments for effectiveness, efficiency and visualization in Appendix C, and the pseudo code of SGOOD in Appendix D.

## A  PROOF FOR PROPOSITION 1

*Proof.* We first prove that SGOOD is at least as powerful as 1&2-WL in Lemma 1. Then, we prove that SGOOD can distinguish 2-regular graphs that 1&2-WL cannot distinguish in Lemma 2. Combining these two Lemmas, we prove that SGOOD is strictly more powerful than 1&2-WL.

□

**Lemma 1.** Let $G_1 = (V_1, E_1)$ and $G_2 = (V_2, E_2)$ be two graphs identified as non-isomorphic by 1&2-WL. SGOOD projects them into different representations $\mathbf{h}_{\mathcal{G}_1}$ and $\mathbf{h}_{\mathcal{G}_2}$ in Eq. (4).

*Proof.* Let $\mathcal{H}_1^G = \{\mathbf{h}_v | v \in V_1\}$ and $\mathcal{H}_2^G = \{\mathbf{h}_v | v \in V_2\}$ be the multisets of node representations of $G_1$ and $G_2$ generated by GIN in Eq. (2), respectively. Let $\mathcal{G}_1 = (\mathcal{V}_1, \mathcal{E}_1)$ and $\mathcal{G}_2 = (\mathcal{V}_2, \mathcal{E}_2)$ be the super graphs of $G_1$ and $G_2$ respectively. We consider two cases: (1) $|\mathcal{V}_1| \neq |\mathcal{V}_2|$, (2) $|\mathcal{V}_1| = |\mathcal{V}_2|$.

For case (1), $\mathcal{G}_1$ and $\mathcal{G}_2$ are two graphs with different number of nodes. Thus, $\mathcal{G}_1$ and $\mathcal{G}_2$ can be easily determined as non-isomorphic by 1&2-WL. It is proved that GIN with sufficient number of layers and all injective functions is as powerful as 1&2-WL (Xu et al., 2018). As GIN is adopted in SGOOD as GNN backbone with sufficient number of layers and READOUT function in Eq.(4) is injective, representations $\mathbf{h}_{\mathcal{G}_1}$ and $\mathbf{h}_{\mathcal{G}_2}$ of $\mathcal{G}_1$ and $\mathcal{G}_2$ are different.

For case (2), let $|\mathcal{V}_1| = |\mathcal{V}_2| = K$. Let $\mathcal{H}_1^{\mathcal{G}} = \{\mathbf{h}_{g_{1,j}}^{(0)} | g_{1,j} \in \mathcal{V}_1\}$ and $\mathcal{H}_2^{\mathcal{G}} = \{\mathbf{h}_{g_{2,j}}^{(0)} | g_{2,j} \in \mathcal{V}_2\}$ be the multisets of initial node representations of $\mathcal{G}_1$ and $\mathcal{G}_2$ calculated by Eq.(2), respectively. Using GIN with sufficient number of layers, we have $\mathcal{H}_1^G \neq \mathcal{H}_2^G$ (Xu et al., 2018). As stated in Section 3.1, the substructures $\{g_{i,j}\}_{j=1}^{n_i}$ of a graph $G_i$ satisfy the following properties: (i) the substructures are non-overlapping, (ii) the union of the nodes in all substructures is the node set of $G_i$. Thus, $\{\{\mathbf{h}_v | v \in g_{1,j}\}\}_{j=1}^K$ (resp. $\{\{\mathbf{h}_v | v \in g_{2,j}\}\}_{j=1}^K$) is a partition of $\mathcal{H}_1^G$ (resp. $\mathcal{H}_2^G$). Then, we have $\{\{\mathbf{h}_v | v \in g_{1,j}\}\}_{j=1}^K \neq \{\{\mathbf{h}_v | v \in g_{2,j}\}\}_{j=1}^K$. As POOL function in Eq.(2) is injective, we have $\{\texttt{POOL}(\{\mathbf{h}_v | v \in g_{1,j}\})\}_{j=1}^K \neq \{\texttt{POOL}(\{\mathbf{h}_v | v \in g_{2,j}\})\}_{j=1}^K$, that is $\mathcal{H}_1^{\mathcal{G}} \neq \mathcal{H}_2^{\mathcal{G}}$. As GIN with sufficient number of layers and READOUT function in Eq.(4) are both injective, we derive that representations $\mathbf{h}_{\mathcal{G}_1}$ and $\mathbf{h}_{\mathcal{G}_2}$ generated on $\mathcal{H}_1^{\mathcal{G}}$ and $\mathcal{H}_2^{\mathcal{G}}$ are different.

Combining case (1) and case (2), we prove Lemma 1.  □

Next, we prove that SGOOD can distinguish 2-regular graphs that 1&2-WL cannot distinguish in Lemma 2. Before that, we first give the definition of 2-regular graphs. Note that we only consider undirected graphs in this paper.

**Definition 2** (2-regular graph). A graph is said to be regular of degree 2 if all local degrees are 2.

Based on the definition of a 2-regular graph, we can conclude that a 2-regular graph consists of one or more (disconnected) cycles.

**Lemma 2.** Given two non-isomorphic $n$-node 2-regular graphs $G_1 = (V_1, E_1)$ and $G_2 = (V_2, E_2)$ that 1&2-WL cannot distinguish, SGOOD projects them into different graph representations $\mathbf{h}_{\mathcal{G}_1}$ and $\mathbf{h}_{\mathcal{G}_2}$ in Eq. (4).

*Proof.* Based on the definition of a 2-regular graph, we can say that $G_1$ and $G_2$ consist of one or more disconnected cycles. Let $r_1$ and $r_2$ be the number of cycles in $G_1$ and $G_2$, respectively. We consider two cases: (1) $r_1 \neq 1 \wedge r_2 \neq 1$, (2) $(r_1 = 1 \wedge r_2 \neq 1) \vee (r_1 \neq 1 \wedge r_2 = 1)$.

For case (1), $G_1$ and $G_2$ consist of disconnected circles. Let $\mathcal{G}_1 = (\mathcal{V}_1, \mathcal{E}_1)$ and $\mathcal{G}_2 = (\mathcal{V}_2, \mathcal{E}_2)$ be the constructed super graphs of $G_1$ and $G_2$, respectively. $\mathcal{G}_1$ and $\mathcal{G}_2$ are constructed by modularity-based community detection method (Clauset et al., 2004) that assign nodes in a graph to different clusters when the modularity of the graph is maximized under such cluster assignment. As Brandes et al.

(2007) proves in Lemma 3.4, there is always a clustering with maximum modularity, in which each cluster consists of a connected subgraph. As a result, $\forall g_{1,j} \in \mathcal{V}_1$ is a circle in $G_1$, and $|\mathcal{V}_1| = r_1$. Similarly, $\forall g_{2,j} \in \mathcal{V}_2$ is a circle in $G_2$, and $|\mathcal{V}_2| = r_2$. Let $\mathcal{N}_1 = \{|V_{1,j}|\}_{j=1}^{|\mathcal{V}_1|}$ and $\mathcal{N}_2 = \{|V_{2,j}|\}_{j=1}^{|\mathcal{V}_2|}$. Since $G_1$ and $G_2$ are non-isomorphic, we have $\exists n_{1,j} \in \mathcal{N}_1 : \forall n_{2,j} \in \mathcal{N}_2, n_{1,j} \neq n_{2,j}$. As a result, we have $\mathcal{N}_1 \neq \mathcal{N}_2$. Then, we have $\{\{\mathbf{h}_v | v \in V_{1,j}\}\}_{j=1}^{|\mathcal{V}_1|} \neq \{\{\mathbf{h}_v | v \in V_{2,j}\}\}_{j=1}^{|\mathcal{V}_2|}$. As POOL function in Eq.(2) is injective, we have $\{\text{POOL}(\{\mathbf{h}_v | v \in g_{1,j}\})\}_{j=1}^{|\mathcal{V}_1|} \neq \{\text{POOL}(\{\mathbf{h}_v | v \in g_{2,j}\})\}_{j=1}^{|\mathcal{V}_2|}$, that is $\mathcal{H}_1^{\mathcal{G}} \neq \mathcal{H}_2^{\mathcal{G}}$. As GIN with sufficient number of layers and READOUT function in Eq.(4) are both injective, we have representations $\mathbf{h}_{\mathcal{G}_1}$ and $\mathbf{h}_{\mathcal{G}_2}$ generated on $\mathcal{H}_1^{\mathcal{G}}$ and $\mathcal{H}_2^{\mathcal{G}}$ are different.

For case (2), we consider $r_1 = 1 \wedge r_2 \neq 1$, and the proof when $r_2 = 1 \wedge r_1 \neq 1$ is similar. $G_1$ consists of one single circle, and $G_2$ consists of $r_2$ disconnected circles. Let $\mathcal{G}_1 = (\mathcal{V}_1, \mathcal{E}_1)$ and $\mathcal{G}_2 = (\mathcal{V}_2, \mathcal{E}_2)$ be the constructed super graphs of $G_1$ and $G_2$, respectively. For $G_2$ and $\mathcal{G}_2$, $\forall g_{2,j} \in \mathcal{V}_2$ is a circle in $G_2$, and $|\mathcal{V}_2| = r_2$ following the conclusion in case (1). For $G_1$ and $\mathcal{G}_1$, we consider two cases: (i) $|\mathcal{V}_1| = r_1 = 1$, and (ii) $|\mathcal{V}_1| > 1$. For case (i), $\mathcal{V}_1 = \{g_{1,1}\}$, where $g_{1,1} = G_1$. Let $\mathcal{N}_1 = \{|V_{1,j}|\}_{j=1}^{|\mathcal{V}_1|} = \{|V_{1,1}|\}$ and $\mathcal{N}_2 = \{|V_{2,j}|\}_{j=1}^{|\mathcal{V}_2|}$, where $|\mathcal{V}_2| > 1$. We have $\mathcal{N}_1 \neq \mathcal{N}_2$. Similar to case (1), we have the same conclusion that graph representations $\mathbf{h}_{\mathcal{G}_1}$ and $\mathbf{h}_{\mathcal{G}_2}$ generated on $\mathcal{H}_1^{\mathcal{G}}$ and $\mathcal{H}_2^{\mathcal{G}}$ are different. For case (ii), $\mathcal{V}_1 = \{g_{1,j}\}_{j=1}^{|\mathcal{V}_1|}$, where $\forall g_{1,j} \in \mathcal{V}_1$ is a chain and two nearby chain are connected in $\mathcal{G}_1$. In other words, $\mathcal{G}_1$ is a $|\mathcal{V}_1|$-circle while $\mathcal{G}_2$ consists of $|\mathcal{V}_2|$ isolated nodes. Thus, $\mathcal{G}_1$ and $\mathcal{G}_2$ can be easily distinguished as non-isomorphic by 1&2-WL. According to (Xu et al., 2018), when we encode $\mathcal{G}_1$ and $\mathcal{G}_2$ by Eq. (3) with sufficient layers of GIN, and generate $\mathbf{h}_{\mathcal{G}_1}$ and $\mathbf{h}_{\mathcal{G}_2}$ by Eq. (4), where READOUT is injective, $\mathbf{h}_{\mathcal{G}_1}$ and $\mathbf{h}_{\mathcal{G}_2}$ are different. Combining case (i) and case (ii), we prove that SGOOD generates different $\mathbf{h}_{\mathcal{G}_1}$ and $\mathbf{h}_{\mathcal{G}_2}$ for $G_1$ and $G_2$ in case (2).

Combining case (1) and case (2), we prove Lemma 2. $\qquad \square$

## B  EXPERIMENTAL SETTINGS

We provide more details on datasets, evaluation metrics, and implementation here for reproducibility. All experiments are conducted on a Linux server with Intel Xeon Gold 6226R 2.90GHz CPU and an Nvidia RTX 3090 GPU card.

### B.1  DATASET DETAILS

We adopt real-world datasets in various data domains for graph-level OOD detection. The dataset statistics is listed in Table 2. Following existing work (Liu et al., 2023b; Li et al., 2022b), given a graph dataset, we use graphs of the same type with distribution shift as ID and OOD data, respectively. The detailed descriptions of ID and OOD graphs in the 6 datasets are as follows.

- **ENZYMES (Morris et al., 2020)** dataset comprises protein networks representing enzymes classified into 6 EC top-level classes. In this paper, we consider graphs from *all* classes in ENZYMES as in-distribution (ID) graphs. To introduce OOD graphs, we utilize graphs from the PROTEINS dataset (Morris et al., 2020). PROTEINS is also a dataset of protein networks, where graphs are labeled as either 'Enzymes' or 'Non-enzymes'. Specifically, we use graphs in PROTEINS with label 'Non-enzymes' as OOD graphs. Consequently, the OOD graphs in ENZYMES represent unseen classes.

- **IMDB-M** (Morris et al., 2020) is a dataset of social networks. It consists of ego-networks derived from actor collaborations. The graphs are labeled with three genres: Comedy, Romance, and Sci-Fi. We consider graphs from *all* classes in IMDB-M as ID graphs. To introduce OOD graphs, we utilize graphs from another dataset called IMDB-B (Morris et al., 2020). Similar to IMDB-M, IMDB-B is also a dataset of social networks, but the graphs are labeled as either 'Action' or 'Romance'. Specifically, we use graphs labeled as 'Action' as OOD graphs. These OOD graphs do not belong to any classes in IMDB-M, and they represent unseen classes.

- **IMDB-B** (Morris et al., 2020) dataset is constructed in a similar manner to IMDB-M. Specifically, we consider graphs from *both* classes (Action and Romance) in IMDB-B as ID graphs. On the other hand, we regard graphs labeled as 'Comedy' or 'Sci-Fi' in IMDB-M as OOD graphs. These OOD graphs represent classes that are not present in IMDB-B, and they are with unseen classes.

- **REDDIT-12K** (Yanardag & Vishwanathan, 2015) dataset is a large-scale dataset of social networks. It consists of graphs corresponding to an online discussion thread in REDDIT where nodes correspond to users. The graphs are labeled as 11 classes based on the subreddit they belong to. In this paper, we consider graphs from *all* classes in REDDIT-12K as in-distribution (ID) graphs. To introduce OOD graphs, we utilize graphs from the REDDIT-BINARY dataset (Yanardag & Vishwanathan, 2015) where graphs also represents online discussion threads and are labeled as question/answer-based community or a discussion-based community. Consequently, the OOD graphs in REDDIT-12K represent unseen classes.

- **BACE** (Wu et al., 2018) is a dataset of molecular graphs used for predicting particular physiology properties of chemical compounds. The dataset is split into training/validation/test sets based on the scaffolds of molecules. Notably, the samples in the training set have distinct scaffolds compared to those in the validation and test sets. The molecular properties of different scaffolds are often quite different (Ji et al., 2022). We consider graphs from the training set as ID graphs, while graphs from the test set are treated as OOD graphs. The OOD graphs exhibit a scaffold distribution that differs from ID graphs.

- **BBBP** (Wu et al., 2018) is a dataset of molecular graphs for predicting barrier permeability. Like BACE dataset, BBBP is split into training, validation, and test sets based on the scaffolds of molecules. We consider graphs from the training set as ID graphs, while graphs from the test set are treated as OOD graphs. The OOD graphs exhibit a scaffold distribution that differs from the ID graphs.

- **DrugOOD** is a dataset of molecular graphs generated by the dataset curator provided by (Ji et al., 2022), which is a systematic OOD dataset curator and benchmark for AI-aided drug discovery. We focus on the sub-dataset DrugOOD-sbap-core-ec50-protein, which contains molecular graphs for the task structure-based affinity prediction. The dataset is split into training/validation/test sets based on the protein target of molecules. Graphs from the training set are considered ID graphs, while graphs from the test set are treated as OOD graphs. The OOD graphs exhibit a protein target distribution that differs from that of the ID graphs.

- **HIV** (Wu et al., 2018) is a large-scale dataset of molecular graphs for testing compounds on the ability to inhibit HIV replication. Like BACE and BBBP dataset, HIV is split into training, validation, and test sets based on the scaffolds of molecules. We consider graphs from the training set as ID graphs, while graphs from the test set are treated as OOD graphs. The OOD graphs exhibit a scaffold distribution that differs from the ID graphs.

## B.2 EVALUATION METRICS

We explain in details the OOD detection evaluation metrics. We use three commonly-used metrics AUROC, AUPR and FPR95 for OOD detection evaluation (Hendrycks & Gimpel, 2017; Wu et al., 2022). All the three metrics are independent of threshold choosing.

- **AUROC**, short for Area Under the Receiver Operating Characteristic (ROC) Curve, is a widely used performance metric. It quantifies the area under the ROC curve, which plots the True Positive Rate (TPR) against the False Positive Rate (FPR) across different probability thresholds ranging from 0 to 1. The AUROC score provides a comprehensive assessment of a model's ability to differentiate between the positive and negative classes, reflecting its overall discriminative power.

- **AUPR** stands for Area Under the Precision-Recall Curve. Precision-Recall curve is a plot of precision versus recall at various probability thresholds ranging from 0 to 1. Higher AUPR indicates that positive samples are correctly identified while false positive predictions are minimized. AUPR is particularly useful in imbalanced datasets where one class is significantly underrepresented compared to the other.

- **FPR95** stands for False Positive Rate at 95% True Positive Rate. FPR95 measures the false positive rate (FPR) when the true positive rate (TPR) is 95%. A lower FPR95 value indicates better performance, as it means the classifier is able to maintain a high true positive rate while minimizing false positive predictions.

## B.3 IMPLEMENTATION DETAILS OF BASELINES

We provide more description and implementation details of the baselines in Section 4.

- **MSP, Energy, ODIN** are general OOD detection methods that estimate OOD scores directly from classification logits at test time. Specially, MSP (Hendrycks & Gimpel, 2017) is the first and the most basic baseline that directly uses the maximum softmax score as OOD score. Energy (Liu et al., 2020) uses energy function that works directly on the output logits to predict OOD scores. ODIN (Liang et al., 2018) uses temperature scaling with gradient-based input perturbations to enlarge the outputs differences between OOD and ID samples.

- **MAH** (Lee et al., 2018) is a distance-based OOD detection method. It models the feature embedding space as a mixture of multivariate Gaussian distributions and measures OOD scores according to the MAH distance between test samples and ID training data.

- **GNNSafe** (Wu et al., 2022) is a graph OOD detection method based on energy model. It incorporates GNNs in the energy model and detects OOD samples using energy scores. For node-level OOD detection, it further adopts a propagation scheme to leverage graph structure through unlabeled nodes. In our paper, we use graph labels to directly run the basic version of GNNSafe from its Section 3.1 (Wu et al., 2022).

- **GraphDE** (Li et al., 2022b) is a graph-level OOD detection method based on probabilistic model. It addresses both the challenges of debiased learning and OOD detection in graph data. By modeling the graph generative process and incorporating a latent environment variable, the model can automatically identify outliers during training and serve as an effective OOD detector.

- **GOOD-D** (Liu et al., 2023b) is an unsupervised graph-level OOD detection method. It detects OOD graphs solely based on unlabeled ID data. GOOD-D utilizes a graph contrastive learning framework combined with perturbation-free graph data augmentation to capture latent ID patterns and detect OOD graphs based on semantic inconsistency at multiple levels of granularity.

- **OCGIN** (Zhao & Akoglu, 2021) is a graph-level anomaly detection method that combines deep one-class classification with GIN (Xu et al., 2018). It aims to project the outlier graphs at a significant distance from the training graphs in the learned feature space.

- **OCGTL** (Qiu et al., 2022) is a graph-level anomaly detection method based on self-supervised learning and transformation learning. It develops an one-class objective that encourages graph embeddings of training data to concentrate within a hyper-sphere and outlier graphs are distant to the hyper-sphere.

- **GLocalKD** (Ma et al., 2022) is a graph-level anomaly detection method. By training a predictor network to reproduce representations from a randomly-initialized network, the model learns both global and local normal patterns in the training data. Anomaly scores are then computed based on the prediction error, allowing the detection of irregular or abnormal graphs.

**Implementation Details.** For MSP, Energy, ODIN, and MAH baselines, we substitute the network backbone in their official implementation with a 5-layer GIN (Xu et al., 2018) using a fixed hidden dimension of 16 to encode graphs into node representations. The node representations from different layers are first concatenated and then aggregated using sum pooling for the final graph representations. Graph representations are sent to linear layer for classification logits. Other experimental settings are the same as SGOOD. For ODIN, as we lack auxiliary OOD data for hyperparameter fine-tuning, we initially explore the temperature values from $1, 10, 100, 1000$ and the perturbation magnitudes from $0, 0.001, 0.002, 0.004$ on all datasets. After experimental tuning, we set the temperature to 10 and the perturbation magnitude to 0.002 consistently achieves competitive performance across all datasets. This configuration is then fixed for further evaluation. For MAH, we leverage the graph representations used for classification to compute the MAH distance, which serves as the estimated OOD scores. However, we do not employ the calibration techniques, such as input pre-processing and feature ensemble in the original paper (Lee et al., 2018). We observed a significant drop in performance when implementing MAH with these techniques. We guess the reason is that the calibration techniques designed for image data are not suitable for graphs. For the other competitors, we use their original codes provided by the respective authors. All competitors are trained using ID training graphs and fine-tuned using ID graphs in validation set.

## C  ADDITIONAL EXPERIMENTS

**Performance on ID graph classification.**    Table 9 reports the performanc on ID graph classification of all methods by Accuracy (ID ACC). We observe that SGOOD achieves best ID ACC on 7/8 datasets, which indicates that leveraging substructures also benefits graph classification. Nevertheless, note that, as mentioned, the main goal of OOD detection is to accurately identify OOD data during testing, while maintaining instead of significantly improving ID ACC.

Table 9: ID graph classification performance measured by ID ACC. All results are reported in percentage % (mean ± std). / indicates that ID ACC is not applicable for unsupervised methods.

| Method | ENZYMES | IMDB-M | IMDB-B | REDDIT-12K | BACE | BBBP | HIV | DrugOOD |
|---|---|---|---|---|---|---|---|---|
| MSP | 37.33±8.73 | 48.27±3.58 | 69.80±5.38 | 48.91±1.06 | 80.83±2.41 | 87.44±2.57 | 96.62±0.39 | 79.20±6.49 |
| Energy | 37.33±8.73 | 48.27±3.58 | 69.80±5.38 | 48.91±1.06 | 80.83±2.41 | 87.44±2.57 | 96.62±0.39 | 79.20±6.49 |
| ODIN | 37.33±8.73 | 48.27±3.58 | 69.80±5.38 | 48.91±1.06 | 80.83±2.41 | 87.44±2.57 | 96.62±0.39 | 79.20±6.49 |
| MAH | 37.33±8.73 | 48.27±3.58 | 69.80±5.38 | 48.91±1.06 | 80.83±2.41 | 87.44±2.57 | 96.62±0.39 | 79.20±6.49 |
| GNNSafe | 17.66±2.71 | 30.13±5.55 | 50.20±2.70 | 27.42±6.62 | 56.69±8.24 | 79.14±11.55 | 96.58±0.34 | 64.40±20.08 |
| GraphDE | 46.00±2.70 | 37.86±5.89 | 69.80±7.05 | 40.68±8.64 | 77.68±3.65 | 88.90±1.00 | 96.20±0.40 | 77.00±6.39 |
| GOOD-D | / | / | / | / | / | / | / | / |
| OCGIN | / | / | / | / | / | / | / | / |
| OCGTL | / | / | / | / | / | / | / | / |
| GLocalKD | / | / | / | / | / | / | / | / |
| SGOOD | **48.66±3.49** | **48.66±2.77** | **71.60±3.00** | **51.82±1.51** | 80.33±2.84 | **89.14±3.44** | **96.66±0.29** | **79.40±3.81** |

Table 10: Performance with different backbones by AUROC (%). **Bold**: best. Underline: runner-up.

| Backbone | Method | ENZYMES | IMDB-M | IMDB-B | BACE | BBBP | DrugOOD |
|---|---|---|---|---|---|---|---|
| GCN | MAH | 70.04 | 71.27 | 53.46 | 72.68 | 54.97 | 66.01 |
| | GraphDE | 61.40 | 68.44 | 29.13 | 53.24 | 52.50 | 56.61 |
| | GOOD-D | 41.96 | 61.71 | 59.53 | 72.52 | 58.91 | 61.79 |
| | OCGIN | 64.35 | 57.46 | 64.08 | 67.54 | 51.23 | 59.30 |
| | SGOOD | **71.26** | **73.52** | **65.91** | **83.42** | **62.76** | **72.52** |
| GraphSage | MAH | 68.07 | 48.06 | 43.63 | 73.60 | 53.88 | 64.55 |
| | GraphDE | 61.37 | **69.65** | 28.28 | 53.24 | 52.50 | 56.66 |
| | GOOD-D | 45.55 | 57.02 | 23.90 | 73.15 | 56.85 | 61.57 |
| | OCGIN | **71.75** | 36.86 | **71.44** | 57.47 | 46.65 | 63.82 |
| | SGOOD | 70.21 | 68.63 | 61.59 | **82.22** | **59.50** | **68.60** |

**Performance under different backbones other than GIN.**    We evaluate the performance of SGOOD and competitors when changing the GIN backbone to GCN (Kipf & Welling, 2017) and GraphSage (Hamilton et al., 2017). Table 10 reports the results. Observe that, with GCN backbone, compared with the baselines, SGOOD consistently achieves the best scores; with GraphSage backbone, SGOOD is the best on BACE, BBP, DrugOOD, and the second best on other datasets. The results validate the versatility/robustness of SGOOD to differnt backbones.

**The effect of augmentation ratio.**    We conduct experiments to study the effect of augmentation ratio on the three substructure-preserving graph augmentations (SD, SG, SS) introduced in Section 3.2. Specially, we fix $\mathcal{T}_0$ as $I$ that indicates no augmentation, set $\mathcal{T}_1$ as one of the three augmentations, and vary the augmentation ratio (dropping ratio/substitution ratio) from $0.1$ to $0.5$. Intuitively, larger augmentation ratio leads to harder contrastive tasks. Figure 3 reports the results, and the dotted red line indicates the performance of SGOOD\CL without any augmentation for calibration. First, for all three substructure-preserving graph augmentations, under most augmentation ratio settings, we can achieve better performance than the red-dot baseline. Second, the three augmentations usually achieve the most significant performance improvement in SGOOD under moderate augmentation ratio (*e.g.*, 0.3 and 0.4).

**The effect of pretraining epochs.**    We conduct experiments to study the effect of pretraining epochs $T_{PT}$ from 0 to 200. As shown in Figure 4, compared to SGOOD without first-stage pretraining ($T_{PT} = 0$), pretraining improves SGOOD's performance. We also found that excessive pretraining can sometimes have negative effects. For example, when $T_{PT} = 200$, SGOOD's performance

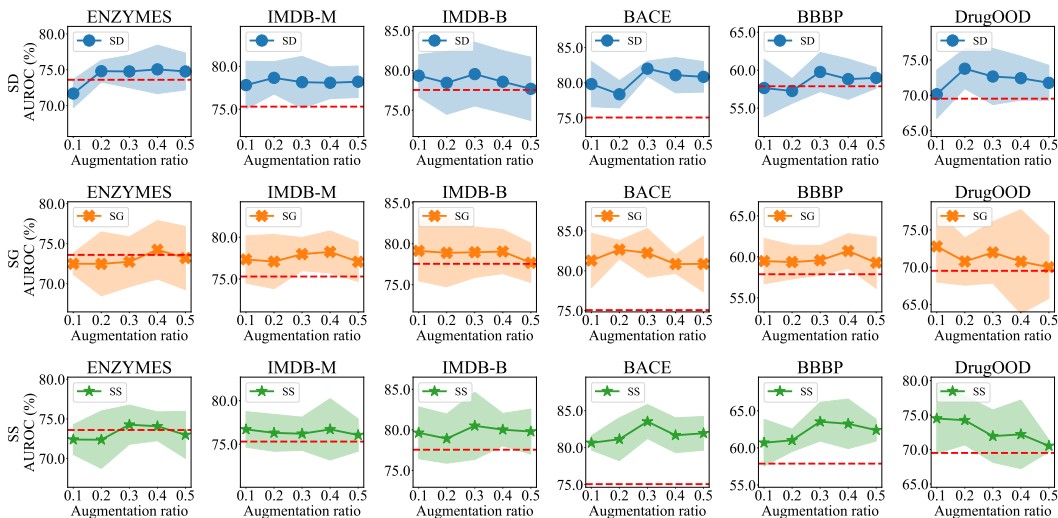

Figure 3: OOD detection performance of SGOOD when augmentation ratio varies by AUROC (%) on all the three substructure-preserving graph augmentations, SD, SG, and SS. The dotted red line indicates the performance of SGOOD\CL without any augmentation, which serves as a base performance. The area in color represents standard deviation.

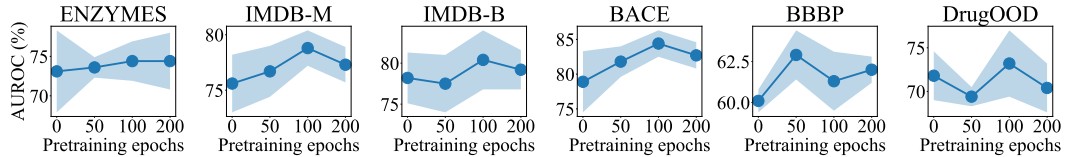

Figure 4: OOD detection performance of SGOOD by AUROC (%) when the number of pretraining epochs $T_{PT}$ varies from 0 to 200, with colored area representing standard deviation.

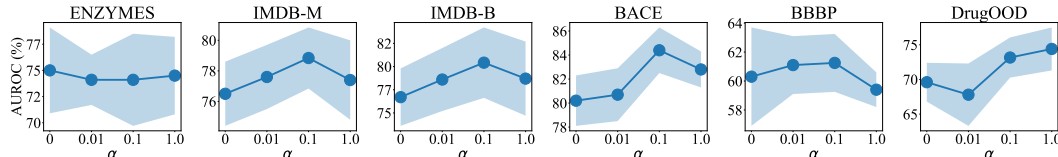

Figure 5: OOD detection results of SGOOD by AUROC (%) when the weight of the contrastive loss $\alpha$ varies from 0 to 1, with colored area representing standard deviation.

decrease on all datasets except ENZYMES. We speculate the reason is that excessive pretraining makes task-agnostic information dominate, with a negative impact on the SGOOD's ability to learn from class labels. As $T_{PT} = 100$ generally leads to competitive performance across all datasets, we set the default value of $T_{PT}$ to 100.

**The effect of $\alpha$.**  We vary the weight of the contrastive loss $\alpha$ from 0 to 1 to study the effect. As shown in Figure 5, compared to SGOOD fine-tuned solely by $\mathcal{L}_{CE}$ (*i.e.*, $\alpha = 0$), fine-tuning SGOOD with both $\mathcal{L}_{CE}$ and $\mathcal{L}_{CL}$ generally leads to better performance. As $\alpha = 0.1$ usually leads to competitive performance across all datasets, we set the default value of $\alpha$ to 0.1 in SGOOD.

**The effect of the number of negative samples in $\mathcal{L}_{CL}$.**  We conduct experiments to study the effect of the number of negative samples used in contrastive loss $\mathcal{L}_{CL}$ (Eq.(6)). Following the established convention in graph contrastive learning You et al. (2020), pairs of augmented graphs originating from the same graph are treated as positive pairs, while pairs generated from different graphs within the batch are considered negative pairs. In such a way, in a $B$-size batch, for every $G_i$,

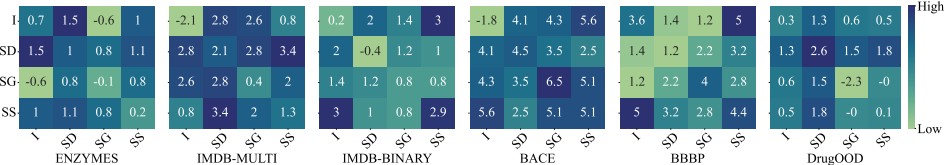

Figure 6: OOD detection results of SGOOD by AUROC (%) when the batch size $B$ varies from 16 to 256.

Figure 7: AUROC gain (%) of SGOOD compared with SGOOD\A without graph augmentations.

Table 11: Comparison of training time per epoch and inference time per epoch of all the methods on six datasets by seconds (s).

| Method | ENZYMES | | IMDB-B | | IMDB-M | | REDDIT-12K | | BACE | | BBBP | | HIV | | DrugOOD | |
|---|---|---|---|---|---|---|---|---|---|---|---|---|---|---|---|---|
| | Train (s) | Test (s) | Train (s) | Test (s) | Train (s) | Test (s) | Train (s) | Test (s) | Train (s) | Test (s) | Train (s) | Test (s) | Train (s) | Test (s) | Train (s) | Test (s) |
| MSP | 0.119 | 0.008 | 0.090 | 0.007 | 0.077 | 0.007 | 0.890 | 0.260 | 0.053 | 0.005 | 0.055 | 0.006 | 2.740 | 0.200 | 0.078 | 0.005 |
| Energy | 0.119 | 0.008 | 0.090 | 0.007 | 0.077 | 0.007 | 0.890 | 0.260 | 0.053 | 0.005 | 0.055 | 0.006 | 2.740 | 0.200 | 0.078 | 0.005 |
| ODIN | 0.119 | 0.026 | 0.090 | 0.022 | 0.077 | 0.021 | 0.890 | 0.420 | 0.053 | 0.016 | 0.055 | 0.013 | 2.740 | 0.300 | 0.078 | 0.020 |
| MAH | 0.119 | 0.020 | 0.090 | 0.019 | 0.077 | 0.018 | 0.890 | 0.400 | 0.053 | 0.016 | 0.055 | 0.019 | 2.740 | 0.300 | 0.078 | 0.019 |
| GraphDE | 1.692 | 0.358 | 1.392 | 0.292 | 1.175 | 0.380 | 176.400 | 0.120 | 0.950 | 0.155 | 0.696 | 0.138 | 43.770 | 9.620 | 1.020 | 0.232 |
| GOOD-D | 0.257 | 0.006 | 0.197 | 0.008 | 0.171 | 0.009 | 17.550 | 0.710 | 0.157 | 0.006 | 0.095 | 0.008 | 5.160 | 0.010 | 0.230 | 0.006 |
| OCGIN | 0.123 | 0.005 | 0.086 | 0.006 | 0.079 | 0.005 | 1.650 | 0.170 | 0.075 | 0.005 | 0.044 | 0.005 | 2.900 | 0.050 | 0.099 | 0.005 |
| GLocalKD | 0.072 | 0.853 | 0.054 | 0.629 | 0.203 | 0.707 | 142.000 | 37.670 | 0.052 | 0.527 | 0.035 | 0.320 | 4.220 | 30.420 | 0.067 | 0.788 |
| SGOOD | 0.161 | 0.030 | 0.137 | 0.027 | 0.138 | 0.028 | 0.980 | 0.130 | 0.085 | 0.025 | 0.058 | 0.028 | 3.970 | 0.300 | 0.124 | 0.030 |

it will have $2B - 2$ negative samples, as shown in the denominator of Eq.(6). Apparently the number of negative samples is related to batch size $B$. We vary $B$ from 16 to 256 to evaluate sensitivity of SGOOD w.r.t. the number of negative samples, and report the results in Figure 6 . Observe that as increasing from 16 to 128, the overall performance increases and then becomes relatively stable, which proves the effectiveness of the augmentation techniques developed in SGOOD and also validates the superior performance of SGOOD when varying batch size and the number of negative samples.

**Visualizing pairwise combinations of all augmentations.** In Figure 7, we exhaust the pairwise combinations of all options in $\mathcal{A} = \{I, SD, SG, SS\}$ and visualize the AUROC gain on graph-level OOD detection over SGOOD\A without graph augmentations. As shown in Figure 7, most combinations achieve positive gains for effective OOD detection.

**Model efficiency.** We compare the training time per epoch and inference time per epoch in seconds of all methods, with results in Table 11. Compared with other graph-level OOD detection competitors, including GraphDE and GOOD-D, SGOOD requires less time to train. Compared with all methods, including the methods originally designed for image data, SGOOD requires moderate time for training. In terms of inference time, SGOOD is much more efficient than GraphDE. Although GOOD-D is more efficient in inference, it is not as accurate as SGOOD in OOD detection as shown in Table 3. Considering together the time cost in Table 11 and the effectiveness in Table 3, we can conclude that SGOOD has superior accuracy for graph-level OOD detection, while being reasonably efficient.

**Substructure Visualization.** In SGOOD, we adopt the modularity-based community detection method (Clauset et al., 2004) to detect substructures in a graph. We demonstrate the detected substructures in different datasets in Figure 8. We observe that dense cliques in protein networks (ENZYMES) and social networks (IMDB-M, IMDB-B) are separated as substructures in SGOOD. For molecular graphs, the rings that play a critical role in the properties of molecules(Zhu et al., 2022) are detected in SGOOD. As the cliques and rings can not be captured by graph representations generated by GNNs based on message passing and flat pooling (Chen et al., 2020) while SGOOD

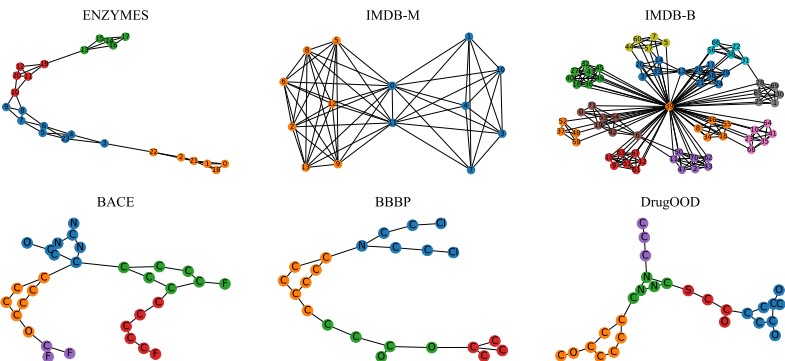

Figure 8: Different colors indicate different substructures. For molecular graphs, we label the nodes with atom types. For graphs of other types, we label the nodes with node IDs.

can generate substructure-enhanced graph representations, it explains why SGOOD achieves superior performance in graph-level OOD detection.

## D   PSEUDO-CODE OF SGOOD

We present the pseudo code of SGOOD for training and testing in Algorithm 1 and 2 respectively.

---

**Algorithm 1:** Pseudo-code of SGOOD (Training)

---

1 **Input:** Training dataset $D_{tr}^{in} = \{(G_i, y_i)\}_{i=1}^n$, testing set $D_{test}$, weight of the contrastive loss $\alpha$, number of first-stage pretraining epoch $T_{PT}$, number of second-stage fine-tuning epoch $T_{FT}$

   // Super graph construction

2 Construct super graphs of substructures $\{\mathcal{G}_i\}_{i=1}^n$ of all $G_i \in D_{tr}^{in}$;

   // First stage

3 **for** $epoch = 1, 2, \ldots, T_{PT}$ **do**

4      Randomly split training graphs $D_{tr}^{in}$ into batches $\mathcal{B}$ with batch size $B$;

5      **for** $\{G_i\}_{i=1}^B \in \mathcal{B}$ **do**

6         **for** $G_i \in \{G_i\}_{i=1}^B$ **do**

7            Obtain augmented super graphs $\widehat{\mathcal{G}}_{i,0}, \widehat{\mathcal{G}}_{i,1}$ by applying $\mathcal{T}_0$ and $\mathcal{T}_1$ to $\mathcal{G}_i$;

8            Obtain augmented graphs $\widehat{G}_{i,0}, \widehat{G}_{i,1}$ according to $\widehat{\mathcal{G}}_{i,0}, \widehat{\mathcal{G}}_{i,1}$;

9            Calculate $\mathbf{h}_{\widehat{\mathcal{G}}_{i,0}}$ and $\mathbf{h}_{\widehat{\mathcal{G}}_{i,1}}$ using $(\widehat{G}_{i,0}, \widehat{\mathcal{G}}_{i,0})$ and $(\widehat{G}_{i,1}, \widehat{\mathcal{G}}_{i,1})$ by Eq.(1)-(4);

10           Obtain $\mathbf{u}_{\widehat{\mathcal{G}}_{i,0}} = \frac{\psi(\mathbf{h}_{\widehat{\mathcal{G}}_{i,0}})}{||\psi(\mathbf{h}_{\widehat{\mathcal{G}}_{i,0}})||}$ and $\mathbf{u}_{\widehat{\mathcal{G}}_{i,1}} = \frac{\psi(\mathbf{h}_{\widehat{\mathcal{G}}_{i,1}})}{||\psi(\mathbf{h}_{\widehat{\mathcal{G}}_{i,1}})||}$ using shared projection head $\psi$ followed by $l_2$-normalization;

11         Calculate contrastive loss $\mathcal{L}_{CL}$ by Eq.(6);

12         Update parameters using mini-batch gradient descent *w.r.t.* $\mathcal{L}_{CL}$;

   // Second stage

13 **for** $epoch = 1, 2, \ldots, T_{FT}$ **do**

14      Randomly split training graphs $D_{tr}^{in}$ into batches $\mathcal{B}$ with batch size $B$;

15      **for** $\{G_i\}_{i=1}^B \in \mathcal{B}$ **do**

16         **for** $G_i \in \{G_i\}_{i=1}^B$ **do**

17           Same as Lines 7-10 ;

18           Calculate $\mathbf{h}_{\mathcal{G}_i}$ using $(G_i, \mathcal{G}_i)$ by Eq.(1)-(4);

19           Calculate prediction logits $\widehat{y}_i$ by applying linear transformation on $\mathbf{h}_{\mathcal{G}_i}$;

20         Calculate cross-entropy loss $\mathcal{L}_{CE}$ by Eq.(5);

21         Calculate contrastive loss $\mathcal{L}_{CL}$ by Eq.(6);

22         Update parameters using mini-batch gradient descent *w.r.t.* $\mathcal{L}_{CE} + \alpha\mathcal{L}_{CL}$;

   // Estimate class centroids and covariance matrix

23 **for** $G_i \in D_{tr}^{in}$ **do**

24      Calculate $\{\mathbf{h}_v | v \in V_i\}$ and $\mathbf{h}_{\mathcal{G}_i}$ using $(G_i, \mathcal{G}_i)$ by Eq.(1)-(4);

25      Calculate $\mathbf{h}_{G_i} = \text{READOUT}(\{\mathbf{h}_v | v \in V_i\})$;

26      Calculate $\mathbf{z}_i = \frac{\text{CONCAT}(\mathbf{h}_{G_i}, \mathbf{h}_{\mathcal{G}_i})}{||\text{CONCAT}(\mathbf{h}_{G_i}, \mathbf{h}_{\mathcal{G}_i})||_2}$;

27 Calculate estimated class centroids $\{\boldsymbol{\mu}_c\}_{c=1}^C$ and covariance matrix $\widehat{\Sigma}$ by Eq.(9);

---

**Algorithm 2:** Pseudo-code of SGOOD (OOD Detection During Testing)

---

1 **Input:** The trained SGOOD model $f$, testing set $D_{test}$, estimated class centroids $\{\boldsymbol{\mu}_c\}_{c=1}^C$ , estimated covariance matrix $\widehat{\Sigma}$

   // Testing stage

2 **for** $G_i \in D_{test}$ **do**

3      Construct super graph $\mathcal{G}_i$;

4      Calculate $\{\mathbf{h}_v | v \in V_i\}$ and $\mathbf{h}_{\mathcal{G}_i}$ using $(G_i, \mathcal{G}_i)$ and $f$ by Eq.(1)-(4);

5      Calculate $\mathbf{h}_{G_i} = \text{READOUT}(\{\mathbf{h}_v | v \in V_i\})$;

6      Calculate $\mathbf{z}_i = \frac{\text{CONCAT}(\mathbf{h}_{G_i}, \mathbf{h}_{\mathcal{G}_i})}{||\text{CONCAT}(\mathbf{h}_{G_i}, \mathbf{h}_{\mathcal{G}_i})||_2}$;

7      Calculate OOD score $S(G_i)$ by Eq.(8);

8      **if** $G_i$ *is not OOD based on* $S(G_i)$ **then**

9         Perform classification on $G_i$ via prediction logits $\widehat{y}_i$ by applying linear transformation on $\mathbf{h}_{\mathcal{G}_i}$;

