# OpenReview forum: "SGOOD: Substructure-enhanced Graph-Level Out-of-Distribution Detection"
_ICLR.cc/2024/Conference — Submitted to ICLR 2024_

### Official Review · Reviewer_3dEG · 2023-10-24

**Soundness:** 2 fair
**Presentation:** 2 fair
**Contribution:** 2 fair
**Rating:** 3
**Confidence:** 4

**Summary:**

Drawing upon the observation of prevalent substructure differences between in-distribution (ID) and out-of-distribution (OOD) graphs, this paper introduces SGOOD, a graph-level OOD detection framework. SGOOD enhances OOD graph detection by incorporating mpre substructure information into ID graph representations. It achieves this through the creation of super graphs of substructures, the implementation of a two-level graph encoding pipeline, and the utilization of three graph augmentation techniques for graph representation. Extensive experiments demonstrate the effectiveness of SGOOD in graph-level OOD detection tasks.

**Strengths:**

1.	The paper presents a well-structured writing.
2.	It incorporates state-of-the-art graph-level OOD detection algorithms in comparative experiments.
3.	The paper explores an intriguing and relatively unexplored research area, emphasizing the importance of graph-level OOD detection.

**Weaknesses:**

1. The motivation to improve graph-level OOD detection by encoding more substructure information into graph representations is unclear.
2. The notion that encoding more substructure information into graph representations will enhance graph-level OOD detection faces skepticism. In practice, theoretically more powerful GNNs often under-perform their 1-WL equivalent counterparts across various graph datasets [1]. This is due to the fact that, in cases where node attributes can function as supplements to structural information, nearly all graphs can be differentiated by 1-WL equivalent GNNs. Substructures do not exist in isolation, and are accompanied by a lot of attribute information. Furthermore, these concerns are verified by the results presented in Table 7. Specifically, more powerful GNNs like NGNN and GNN-AK+ fail to outperform 1-WL equivalent GNNs SAG, TopK, and DiffPool in the graph-level OOD detection task.
3. This paper lacks a clear definition of the graph distribution, and it does not explore the factors contributing to the distribution differences between ID and OOD graphs. It places excessive emphasis on the influence of substructures in graph-level OOD detection while neglecting the discussion of node attributes. Two graphs with identical structures but distinct node features may exhibit entirely different distributions.
4. The paper does not explicitly delineate the specific contributions of the proposed method, SGOOD, to the graph-level OOD detection task. Given the existence of many  theoretically more powerful GNNs, it remains unclear why SGOOD better than those GNNs in the graph-level OOD detection task. SGOOD appears to resemble a new GNN with powerful expressiveness rather than a specialized GNN that can identify OOD graphs.
5. Author wrote: "For augmentations, intuitively, if more information about training ID data is preserved, it is easier to distinguish unseen OOD data. The substructure-preserving graph augmentations are designed to achieve this. " Please provide further explanation for “more information”. What we need to do is to embed all the information related to the substructure into the graph representation? In [2], authors proposed that encoding the task-agnostic (e.g., graph classification task-agnostic) information into representations can improve the OOD detection task.

[1] Dwivedi  et al. Benchmarking graph neural networks. arXiv, 2020

[2] Winkens et al. Contrastive training for improved out-of-distribution detection. arXiv, 2020.

**Questions:**

1. Table 1 lacks clarity, making it difficult for readers realize the ID and OOD graphs used in statistics, and these statistical findings rely on prior knowledge.
2. GNNsafe appears to be primarily designed for node-level OOD detection. How can it be implemented at the graph-level?
3. OCGIN, OCGTL, and GLocalKD are predominantly designed for graph anomaly detection, and their use as comparison algorithms may not be entirely appropriate for graph-level OOD detection.
4. Figure 1 does not effectively convey how SGOOD is specifically tailored for the graph-level OOD detection task.

---

> ### Author Response · Authors · 2023-11-16
> **Response to Reviewer 3dEG (1/3) and look forward to your reply**
>
> We thank your effort to review our paper and we appreciate your recognition on the strengths of our paper. Please find our detailed responses below.
>
>
>
> **W1:** The motivation to improve graph-level OOD detection by encoding more substructure information into graph representations is unclear.
> **Q1:** Table 1 lacks clarity, making it difficult for readers realize the ID and OOD graphs used in statistics, and these statistical findings rely on prior knowledge.
> **Response:**
> The motivation of encoding task-agnostic substructures to improve graph-level OOD detection is empirically justified in Table 1 of the paper. The intuition is that substructure differences commonly exist in real-world ID and OOD graphs, and naturally, if the learned representations can encode substructures, which means better differentiation between ID and OOD graphs, the performance of graph-level OOD detection could be improved.
>
> We agree that Table 1 itself is not self-contained, due to the lack of space in introduction, and we have revised Section 1 in the latest version uploaded. Specifically, please refer to Table 2 in Section 4 for ID and OOD graph statistics. Table 1 shows the percentage of OOD test graphs containing substructures never appeared in ID training graphs in each dataset. These percentage values are high, higher than 44% in 4/6 datasets, indicating that many OOD graphs contain substructures rarely appear in ID graphs. This observation motivates our design to encode substructures to improve graph-level OOD detection. Note that the only purpose of this statistical finding is to motivate our idea, and it is not used in the training, since OOD graphs are only available at test time.
>
>
> **W2:** The notion that encoding more substructure information into graph representations will enhance graph-level OOD detection faces skepticism. In practice, theoretically more powerful GNNs often under-perform their 1-WL equivalent counterparts across various graph datasets [1]. This is due to the fact that, in cases where node attributes can function as supplements to structural information, nearly all graphs can be differentiated by 1-WL equivalent GNNs. Substructures do not exist in isolation, and are accompanied by a lot of attribute information. Furthermore, these concerns are verified by the results presented in Table 7. Specifically, more powerful GNNs like NGNN and GNN-AK+ fail to outperform 1-WL equivalent GNNs SAG, TopK, and DiffPool in the graph-level OOD detection task.
> **W4:** The paper does not explicitly delineate the specific contributions of the proposed method, SGOOD, to the graph-level OOD detection task. Given the existence of many theoretically more powerful GNNs, it remains unclear why SGOOD better than those GNNs in the graph-level OOD detection task. SGOOD appears to resemble a new GNN with powerful expressiveness rather than a specialized GNN that can identify OOD graphs.
> **Response:**
> To answer **W2**, we echo your insightful comment in **W5**: "In [2], authors proposed that encoding the task-agnostic (e.g., graph classification task-agnostic) information into representations can improve the OOD detection task." Existing GNNs like NGNN and GNN-AK+ learns task-specific substructures. As mentioned in Section 1, these methods are trained with a focus on classification-related structures. *On the contrary, our method SGOOD preserves graph classification task-agnostic substructures*, so that the generated representations by SGOOD can better distinguish ID and OOD graphs at test time. The input substructures used in SGOOD are not restricted by ID graph labels, and these substructures are extracted by existing methods as a pre-step. This explains the performance improvements by SGOOD over NGNN and GNN-AK+ in Table 7. In order to make the paper complete, the analysis in Section 3.5 just serves as a complementary justification for SGOOD from the perspective of 1&2-WL test.
>
> Then for **W4**, the main contributions of SGOOD are *how to effectively encode task-agnostic substructures into the generated representations* to detect OOD graphs, different from GNNs like NGNN and GNN-AK+ that are to learn task-specific substructures. To achieve this, as shown in Figure 1, we first design a super graph of substructures, which models the relationships between task-agnostic substructures in a graph, and then generate substructure-enhanced graph representations via two-level graph encoding (Section 3.1); we then design substructure-preserving graph augmentations to enrich training data, while keeping task-agnostic substructures themselves intact (Section 3.2); lastly we design proper objective functions in Section 3.3. These techniques in SGOOD are designed with the purpose to encode task-agonistic substructures into representations for better graph-level OOD detection.
>
> We have revised Section 1 and 3 accordingly in the latest version of our paper that has been uploaded.

---

> > ### Author Response · Authors · 2023-11-16
> > **Response to Reviewer 3dEG (2/3)**
> >
> > **W3:** This paper lacks a clear definition of the graph distribution, and it does not explore the factors contributing to the distribution differences between ID and OOD graphs. It places excessive emphasis on the influence of substructures in graph-level OOD detection while neglecting the discussion of node attributes. Two graphs with identical structures but distinct node features may exhibit entirely different distributions.
> > **Response:**
> > Note that different datasets may have different ID and OOD graph distributions. Moreover, the distribution differences (i.e., distribution shifts) between ID and OOD graphs also vary on different datasets. A method dedicated to one distribution shift may not be generalizable to another kind of distribution shift. Therefore, by following the literature in OOD detection, we do not restrict a proposed method to a specific type of graph distributions or a specific type of distribution shifts. As shown in Table 2, for protein networks (ENZYMES) and social networks (IMDB-M, IMDB-B, REDDIT-12K), graph distributions are class distributions [3], and the distribution shifts of OOD graphs are unseen classes, i.e., a test graph is OOD if it belongs to an unseen class. For molecular graphs (BACE, BBBP, DrugOOD, HIV) in Table 2, we follow [4] and regard graph distributions as either scaffold distributions or protein target distributions. It means that OOD molecular graphs are the graphs with their scaffold or protein targets different from ID ones. More details can be found in Appendix B.1.
> >
> > Then, as shown in Table 1, regardless of the types of distribution shifts, substructure differences between OOD and ID graphs commonly exist on various real datasets, and thus, we propose SGOOD that encodes substructures for graph-level OOD detection. We do not restrict SGOOD to a specific type of distribution shift. We agree that it is promising to explore the factors contributing to the distribution differences, and leave it as future work.
> >
> > Moreover, we kindly clarify that SGOOD encodes node attributes together with graph structure using GNNs, as shown in Eq. (1). As a result, two graphs with identical structures, but distinct node features can also be differentiated by SGOOD. In this work, we focus on substructure patterns instead of node attributes.
> >
> >
> >
> >
> > **W5:** Author wrote: "For augmentations, intuitively, if more information about training ID data is preserved, it is easier to distinguish unseen OOD data. The substructure-preserving graph augmentations are designed to achieve this. " Please provide further explanation for “more information”. What we need to do is to embed all the information related to the substructure into the graph representation? In [2], authors proposed that encoding the task-agnostic (e.g., graph classification task-agnostic) information into representations can improve the OOD detection task.
> > **Response:**
> > We thank the reviewer for the insightful comment by mentioning [2]. The substructures used in our method SGOOD are indeed *task-agnostic*, as they are detected by existing methods, before the training of SGOOD. SGOOD encodes these substructures into the generated representations to improve graph-level OOD detection task.
> >
> > Augmentation techniques are usually designed to enrich training data to train models. Therefore, the "more information" here is to just intuitively indicate that the proposed augmentation techniques can help generate more substructure-preserving training samples, and subsequently, we can sufficiently train our method to generate representations that can better distinguish OOD from ID graphs, which is validated in the ablation study in Table 4.

---

> > > ### Author Response · Authors · 2023-11-16
> > > **Response to Reviewer 3dEG (3/3)**
> > >
> > > **Q2:** GNNsafe appears to be primarily designed for node-level OOD detection. How can it be implemented at the graph-level?
> > > **Response:**
> > > GNNSafe is primarily designed for node-level OOD detection. Therefore, we compare with a modified version of GNNSafe, as stated in Appendix B.3. Specifically, in Section 3.2 of the GNNSafe paper, in a single graph, GNNSafe propagates node energy scores that are obtained by training GNN on *node labels*, which however are not available in our graph-level setting where only *graph-level labels* are available. Therefore, we compared with a version of GNNSafe without node energy score propagation.
> > >
> > >
> > > **Q3:** OCGIN, OCGTL, and GLocalKD are predominantly designed for graph anomaly detection, and their use as comparison algorithms may not be entirely appropriate for graph-level OOD detection.
> > > **Response:**
> > > We follow previous studies on graph-level OOD detection [5, 6], and included the comparison to these methods, in order to comprehensively evaluate the performance of our method and existing methods.
> > >
> > >
> > > **Q4:** Figure 1 does not effectively convey how SGOOD is specifically tailored for the graph-level OOD detection task.
> > > **Response:**  Thank you for the suggestion. We have updated Figure 1 and its caption in the latest version of our paper.
> > >
> > >
> > > [1] Dwivedi et al. Benchmarking graph neural networks. arXiv, 2020
> > > [2] Contrastive training for improved out-of-distribution detection.
> > > [3] OpenOOD: Benchmarking Generalized Out-of-Distribution Detection, NIPS'22.
> > > [4] Drugood: Out-of-distribution (ood) dataset curator and benchmark for ai-aided drug discovery–a focus on affinity prediction problems with noise annotations.
> > > [5] GOOD-D: On Unsupervised Graph Out-Of-Distribution Detection, WSDM’23.
> > > [6] GraphDE: A Generative Framework for Debiased Learning and Out-of-Distribution Detection on Graphs

---

> ### Author Response · Authors · 2023-11-21
> **Reminder for discussion.**
>
> Dear Reviewer 3dEG,
>
> We thank your constructive comments, which are solvable. In our responses, we have addressed all your comments and improved our paper. This is a reminder to discuss. Your feedback on our responses is important to us.
>
> Summary of changes:
> - (W1, Q1) We have revised Section 1 to make clearer the motivation of encoding substructures for graph-level OOD detection.
> - (W2, W4, W5) We have revised our paper to highlight that our method SGOOD preserves task-agnostic substructures, and thus better distinguish ID and OOD graphs at test time, compared with existing powerful GNNs, and further explained our technical contributions.
> - (W3) We have clarified that our method does not require or leverage assumptions on graph distributions, and thus is not constrained to a specific underlying graph distribution.  Different types of datasets may have different ID and OOD graph distributions, and SGOOD can versatilely perform superior on all datasets.
> - (Q2, Q3) We have explained how and why to compare with baselines GNNSafe, OCGIN, OCGTL, and GLocalKD.
> - (Q4) As suggested, we have improved Figure 1 and its description.
>
> Best,
> Authors

---

> > ### Comment · Reviewer_3dEG · 2023-11-22
> >
> > Thank you for addressing my concerns. The response has clarified part of my concerns. However, upon reviewing the revised manuscript, I noted significant modifications to the introduction. The current version proposes encoding task-agnostic substructures in the ID graph to improve OOD graph detection—a concept absent in the initial manuscript. This introduction of new elements has given rise to fresh uncertainties for me.
> >
> > Specifically, I question the appropriateness of defining modularity-based substructures as "task-agnostic." The author asserts that these substructures are task-agnostic due to their independence from specific learning tasks, such as graph classification.
> > The assertion may somewhat inaccurate, given that these structures are closely tied to community detection. For instance, graphs within the same class may exhibit highly similar community structures, especially in social networks. The relevance of modularity-based substructures to the graph classification task appears uncertain and contingent upon the specific dataset used.

---

> ### Author Response · Authors · 2023-11-22
> **Further Responses to Reviewer 3dEG**
>
> Dear reviewer 3dEG,
>
> Please find our further clarifications below. Thank you and look forward to your reply.
>
> > Significant changes in Introduction
>
> **Response:** We clarify that the change in Introduction is **just to swap and reorganize the content** in the two paragraphs in blue, which is not significant, if you compare it with the original version. In particular, we adopted your suggestions in W1, Q1, and W5, and reorganized the content to make our motivation clearer. **The change does not affect our technical designs.** We just color the paragraphs for your easy reading.
>
> > The highlight of “task-agnostic”
>
> **Response:** In the original manuscript, we have discussed in Section 1 that “These methods are trained using ID graphs and classification loss with a focus on *classification-related structures* of ID graphs.”, which indicates that these existing methods are task-specific, echoing your comment in W5, W2. On the other hand, our method is indeed using task-agnostic substructures, as indicated by “our framework SGOOD is *orthogonal* to existing subgraph detection methods” in Section 3.1.
> Moreover, following your suggestion, we think that it is better to highlight the term "task-agnostic" in the revision, compared with the original version. Note that **this change also does not affect our technical designs**.
>
> > The author asserts that these substructures are task-agnostic due to their independence from specific learning tasks, such as graph classification. The assertion may somewhat inaccurate, given that these structures are closely tied to community detection.
>
> **Response:** Note that (i) the substructures are identified in a *preprocessing* step without knowing the classification task, and (ii) in the training stage, the substructures are already fixed as input, and *we do not modify them based on class labels*, and thus they are task-agnostic. (iii) We agree that some hidden correlation may exist between class labels and community structures. Therefore, in **Table 6**, besides modularity-based community structures, we have tried *different substructures* detected by different methods, which all improve the performance, compared with the base method without substructures. This validates that **our method is not limited to a certain type of relationship between substructures and class labels.**

---

### Official Review · Reviewer_NoSM · 2023-10-26

**Soundness:** 3 good
**Presentation:** 3 good
**Contribution:** 2 fair
**Rating:** 5
**Confidence:** 4

**Summary:**

This paper proposes novel graph-level OOD detection framework that generates substructure-enhanced representations and uses substructure-preserving graph augmentations for contrastive training.

**Strengths:**

1. The proposed SGOOD outperforms a number of existing baselines.
2. The design of substructure-enhanced representation learning and augmentation is interesting.
3. The paper is well-organized and clear.

**Weaknesses:**

1. The proposed substructure learning on graphs is related to identifying and learning causally invariant substructures, which has been studied in some previous works [1-3].
2. As for Substructure-Preserving Graph Augmentations, although it perserves substructures, it might change the semantics of graphs.


[1] Learning Causally Invariant Representations for Out-of-Distribution Generalization on Graphs
[2] RIGNN: A Rationale Perspective for Semi-supervised Open-world Graph Classification
[3] Debiasing graph neural networks via learning disentangled causal substructure

**Questions:**

How can the proposed SGOOD ensure semantically meaningful substrctures extracted by predefined methods? Why not using other learning based techniques like hypergraph learning, graph pooling or causal learning to extract substructures?

---

> ### Author Response · Authors · 2023-11-16
> **Response to Reviewer NoSM and look forward to your reply**
>
> We thank your effort to review our paper, and appreciate your recognition on (i) the superior performance of our method SGOOD, (ii) the interesting design of our techniques, and (iii) the clarity of the paper. Please find our detailed responses below.
>
> **W1:** The proposed substructure learning on graphs is related to identifying and learning causally invariant substructures, which has been studied in some previous works.
> **Response:**
> We clarify that we do not learn causally invariant substructures. The substructures used in SGOOD are *task-agnostic*, while causally invariant substructures in existing studies are learned with specific tasks. (i) As stated in Section 3.1 (last paragraph on Page 3), our method SGOOD is orthogonal to existing subgraph detection methods, and it is not our focus on how to identify substructures.  (ii) As shown in Figure 1, SGOOD is not relevant to learning causally invariant substructures. We use substructures to enhance the generated representations for graph-level OOD detection. (iii) As shown in Table 6, SGOOD can work with different substructures identified by different methods, and improve performance compared with the base method without substructures.
>
>
> **W2:** As for Substructure-Preserving Graph Augmentations, although it preserves substructures, it might change the semantics of graphs.
> **Response:**
> We highlight that (i) for the purpose of enriching the input graph data, augmentation techniques, including ours, will change the semantics of graphs, but (ii) there is no need to worry about this in SGOOD, since in the training of SGOOD for loss $\mathcal{L}_{CE}$ in Eq. (5), all original input graphs are used *without* semantic changes. Moreover, as shown in Table 4 of the paper, the proposed augmentation techniques indeed bring performance improvements.
>
>
> **Q1:** How can the proposed SGOOD ensure semantically meaningful substructures extracted by predefined methods?
> **Response:**
> We clarify that it is not our focus on how to extract semantically meaningful substructures, as stated in Section 3.1. The substructures are task-agnostic and extracted as a pre-step. Our main goal is how to encode the substructures into effective representations for graph OOD detection, as illustrated in Figure 1. Our design is motivated by the observation in Table 1 that ID and OOD graphs often have different task-agnostic substructures. As validated in Table 6, SGOOD with different substructures by different methods can always improve performance, compared with the base method without substructures.
>
> **Q2:** Why not using other learning based techniques like hypergraph learning, graph pooling or causal learning to extract substructures?
> **Response:**
> (i) In Table 7 of the paper, we have compared with graph pooling methods (SAG, TopK, DiffPool), which is outperformed for graph-level OOD detection. (ii) The focus of SGOOD is to learn expressive representations to encode task-agnostic substructures to distinguish OOD graphs, while it is not our focus on how to extract substructures, as stated in Section 3.1, and it is a different topic from our method. We agree that it is possible to further investigate the impact of learning techniques as future work.

---

> ### Author Response · Authors · 2023-11-22
> **Reminder for discussion.**
>
> Dear Reviewer NoSM,
>
> This is a reminder for discussion. We appreciate your recognition on the superior performance and interesting design of our method.
> Your comments are insightful and solvable. We have further clarified our technical designs and explained experimental results to address all your comments.
>
> Best,
> Authors

---

### Official Review · Reviewer_AhYj · 2023-10-31

**Soundness:** 2 fair
**Presentation:** 3 good
**Contribution:** 2 fair
**Rating:** 5
**Confidence:** 4

**Summary:**

The paper studies out-of-distribution detection on graph data, which is an under-explored research area in GNNs. The authors propose to exploit the substructure information that is invariant between in-distribution and out-of-distribution to endow the model with the OOD detection capabilities. To this end, the authors resort to constructing a super graph of substructures, augmentation for graph data and contrastive loss designs. Experiments with comparison with several SOTA models verify the effectiveness of the model.

**Strengths:**

1. The proposed method seems novel and reasonable

2. The paper is well written and clearly presented

3. The experiment results are strong given the comparison with several SOTA methods

**Weaknesses:**

1. The proposed method seems incremental and redundant

2. Some of the claims are inproperly stated without justification

3. Theoretical contributions are weak

**Questions:**

1. How is the model sensitive to different substructures as prior information? And how does this impact different tasks and datasets?

2. How are the negative samples for contrastive loss constructed? How is the sensitivity of the model w.r.t. number of negative samples?

3. The authors mentioned that GNNSafe [1], which is the state-of-the-art model for out-of-distribution detection on graphs, cannot be directly compared, can it be stated more clear why GNNSafe is not comparable with the methods in the experiment?

4. The experimental datasets already used are small. How does the model perform on large datasets? What is the computation cost compared with others?

[1] Qitian Wu et al., Energy-based out-of-distribution detection for graph neural networks. International Conference on Learning Representations, 2023.

---

> ### Author Response · Authors · 2023-11-16
> **Response to Reviewer AhYj (1/2),  and look forward to your reply**
>
> **Response to Strengths and Weaknesses:** We appreciate your recognition on (1) the novelty and reasonableness of our method, (2) the clarity of our paper, and (3) the strong experimental results compared with existing methods. We clarify that the main focus of this paper is to develop a new method SGOOD that is effective to handle graph-level OOD detection on various real datasets. The analysis in the paper is to justify the effectiveness of our method, while theoretical contribution is not a major focus of this paper. In terms of the first two weakness points, since they are quite general, we focus on addressing your Questions. Please see our responses below.
>
>
> **Q1:** How is the model sensitive to different substructures as prior information? And how does this impact different tasks and datasets?
> **Response:**
> As stated in Section 3.1, our framework SGOOD is orthogonal to existing subgraph detection methods. The substructures used in SGOOD are task-agnostic. As reported in Table 6 of the paper, compared with the base version without substructures (w.o. substructures), SGOOD can always improve OOD detection performance when adopting different substructures identified by different methods (Modularity, Graclus, LP, BRICS). This demonstrates that SGOOD is not sensitive to specific substructures as prior knowledge. Moreover, it is a natural observation that on different datasets, the performance may vary, but as reported in Table 3 of the paper, SGOOD consistently achieves superior performance under various metrics, including AUROC, AUPR, and FRP95, across 8 real datasets.
>
> **Q2:** How are the negative samples for contrastive loss constructed? How is the sensitivity of the model w.r.t. number of negative samples?
> **Response:**
> As stated in Eq. (6) and the paragraph above Eq. (6) in Section 3.3, given a batch with $B$ training graphs, for each training graph $G_i$ with its super graph $\mathcal{G}_i$, we first choose two augmentations $\mathcal{T}_0$ and $\mathcal{T}_1$ among {I, SD, SG, SS} developed in Section 3.2 based on validation, apply augmentations $\mathcal{T}_0$ and $\mathcal{T}_1$ over super graph $\mathcal{G}_i$ to get $\hat{\mathcal{G}} _{i,0}$ and $\hat{\mathcal{G}} _{i,1}$ respectively, and then transform graph $G_i$ accordingly to construct $\hat{G} _{i,0}$ and $\hat{G} _{i,1}$, which are the two augmented samples generated from $G_i$. Following the established convention in graph contrastive learning [1], pairs of augmented graphs originating from the same graph are treated as positive pairs, while pairs generated from different graphs within the batch are considered negative pairs. In such a way, in a $B$-size batch, for every $G_i$, it will have $2B-2$ negative samples, as shown in the denominator of Eq. (6).
> Apparently the number of negative samples is related to batch size $B$. We vary $B$ from 16 to 256 to evaluate sensitivity of SGOOD w.r.t. the number of negative samples, and report the results in Table 1 below. Observe that as increasing from 16 to 128, the overall performance increases and then becomes relatively stable, which proves the effectiveness of the augmentation techniques developed in SGOOD and also validates the superior performance of SGOOD when varying batch size and the number of negative samples. We have added this experiment in Appendix C.
>
> **Table 1.** Varying batch size and the number of negative samples (AUROC)
> |B|ENZYMES|IMDB-M|IMDB-B|BACE|BBBP|DrugOOD|
> |-|-|-|-|-|-|-|
> |16|73.00|77.81|75.44|75.32|59.84|70.50|
> |32|73.91|77.35|78.11|76.99|60.31|71.20|
> |64|74.52|78.13|78.57|80.91|61.61|71.65|
> |128|74.41|78.84|80.42|84.39|61.25|73.15|
> |256|74.41|77.12|79.55|82.43|62.50|71.11|
>
> **Q3:** The authors mentioned that GNNSafe [2], which is the state-of-the-art model for out-of-distribution detection on graphs, cannot be directly compared, can it be stated more clear why GNNSafe is not comparable with the methods in the experiment?
> **Response:**
> We clarify that (i) GNNSafe is compared in experiments, as reported in Table 3, and (ii) the compared GNNSafe is a modified version, since GNNSafe itself is for node-level OOD detection, but not for graph level. Specifically, in Section 3.2 of the GNNSafe paper [2], in a single graph, GNNSafe propagates node energy scores that are obtained by training GNN on *node labels*, which however are not available in our graph-level setting where only *graph-level labels* are available. Therefore, we compared with a modified version of GNNSafe without such node energy score propagation in Table 3 of our paper.

---

> > ### Author Response · Authors · 2023-11-16
> > **Response to Reviewer AhYj (2/2)**
> >
> > **Q4:** The experimental datasets already used are small. How does the model perform on large datasets? What is the computation cost compared with others?
> > **Response:**
> > First, we want to explain that, compared with the existing papers on graph OOD detection [3,4], we are already using many datasets with sufficiently large sizes in experiments, such as HIV with in total *36192* graphs and REDDIT with 9622 graphs in *11* classes, as shown in Table 2 of the paper.
> >
> > Second, the training and inference time of our method SGOOD and baselines have been reported in Table 11 of Appendix C in the submission. Compared with existing methods, SGOOD achieves better or comparable efficiency. Considering together the effectiveness reported in Table 3, we conclude that SGOOD has superior accuracy for graph-level OOD detection, while being reasonably efficient.
> >
> >
> >
> > [1] Graph Contrastive Learning with Augmentations, NIPS'20.
> > [2] Energy-based out-of-distribution detection for graph neural networks, ICLR'23.
> > [3] GOOD-D: On Unsupervised Graph Out-Of-Distribution Detection, WSDM’23.
> > [4] GraphDE: A Generative Framework for Debiased Learning and Out-of-Distribution Detection on Graphs.

---

> ### Author Response · Authors · 2023-11-22
> **Reminder for discussion.**
>
> Dear Reviewer AhYj,
>
> This is a reminder for discussion. We appreciate your recognition of the novelty and reasonableness of our method, the strong experimental results, and the clarity of the paper.
> Your insightful comments are solvable. We have added new experiments and provided clarifications and justifications, to address all your comments.
>
> Best,
> Authors

---

### Meta-Review · Area_Chair_va9j · 2023-12-06

**Metareview:**

This work presents SGOOD, a framework for out-of-distribution (OOD) detection at the graph level, which leverages the distinct connected induced subgraph (CIS) variations observed between in-distribution (ID) and OOD graphs. By integrating more subgraph information into the representations of ID graphs, SGOOD significantly improves the detection of OOD graphs.

- It is interesting that this work uses CIS's for OOD detection while other works (on invariant OOD representations) use CIS's to find invariant graph features that will allow the graphs to work OOD. My guess is that subgraphs can be used both ways depending on the definition of OOD.

- The main sticking point I found in the reviews/discussion seems to arise from the fact that OOD is never formally defined in the paper. OOD is often defined as a relative concept: The graph/label distribution $p(G,y)$ is OOD with respect to classifier $g$. Looking at subgraph structure distributions and declaring a graph to be OOD is confusing, since the graph representation may be invariant to those changes. In the rebuttal the authors described their approach as "task-agnostic". That is, again, a strange definition of OOD. Say, the graph label $y$ is the number of nodes in the graph. Then, there is no OOD issue if the representation is able to count the number of nodes.

**Justification For Why Not Higher Score:**

One cannot write an OOD paper without formally defining what OOD means in their context.

**Justification For Why Not Lower Score:**

N/A

---

### Decision · Program_Chairs · 2024-01-16

Reject